# $N^6$-methyladenosine modification of HIV-1 RNA suppresses type-I interferon induction in differentiated monocytic cells and primary macrophages

**Shuliang Chen**[1◎¤a], **Sameer Kumar**[2◎], **Constanza E. Espada**[2◎], **Nagaraja Tirumuru**[1¤b], **Michael P. Cahill**[2], **Lulu Hu**[3¤c], **Chuan He**[3,4], **Li Wu**[2]*

**1** Center for Retrovirus Research, Department of Veterinary Biosciences, The Ohio State University, Columbus, Ohio, United States of America, **2** Department of Microbiology and Immunology, Carver College of Medicine, University of Iowa, Iowa City, Iowa, United States of America, **3** Department of Chemistry, Department of Biochemistry and Molecular Biology, Institute for Biophysical Dynamics, University of Chicago, Chicago, Illinois, United States of America, **4** Howard Hughes Medical Institute, University of Chicago, Chicago, Illinois, United States of America

◎ These authors contributed equally to this work.
¤a Current address: School of Basic Medical Sciences, Wuhan University, Wuhan, Hubei, China
¤b Current address: Reckitt Benckiser LLC, Parsippany, New Jersey, United States of America
¤c Current address: Institutes of Biomedical Sciences, Fudan University, Shanghai, China
* li-wu@uiowa.edu

**Data Availability Statement:** All relevant data are within the manuscript.

## Abstract

$N^6$-methyladenosine ($m^6A$) is a prevalent RNA modification that plays a key role in regulating eukaryotic cellular mRNA functions. RNA $m^6A$ modification is regulated by two groups of cellular proteins, writers and erasers that add or remove $m^6A$, respectively. HIV-1 RNA contains $m^6A$ modifications that modulate viral infection and gene expression in $CD4^+$ T cells. However, it remains unclear whether $m^6A$ modifications of HIV-1 RNA modulate innate immune responses in myeloid cells that are important for antiviral immunity. Here we show that $m^6A$ modification of HIV-1 RNA suppresses the expression of antiviral cytokine type-I interferon (IFN-I) in differentiated human monocytic cells and primary monocyte-derived macrophages. Transfection of differentiated monocytic U937 cells with HIV-1 RNA fragments containing a single $m^6A$-modification significantly reduced IFN-I mRNA expression relative to their unmodified RNA counterparts. We generated HIV-1 with altered $m^6A$ levels of RNA by manipulating the expression of the $m^6A$ erasers (FTO and ALKBH5) or pharmacological inhibition of $m^6A$ addition in virus-producing cells, or by treating HIV-1 RNA with recombinant FTO *in vitro*. HIV-1 RNA transfection or viral infection of differentiated U937 cells and primary macrophages demonstrated that HIV-1 RNA with decreased $m^6A$ levels enhanced IFN-I expression, whereas HIV-1 RNA with increased $m^6A$ modifications had opposite effects. Our mechanistic studies indicated that $m^6A$ of HIV-1 RNA escaped retinoic acid-induced gene I (RIG-I)-mediated RNA sensing and activation of the transcription factors IRF3 and IRF7 that drive IFN-I gene expression. Together, these findings suggest that $m^6A$ modifications of HIV-1 RNA evade innate immune sensing in myeloid cells.

**Funding:** This work was supported in part by National Institutes of Health grants R01AI150343 and R01AI141495 (to L.W.), and R01 ES030546 (to C.H.). C.H. is a Howard Hughes Medical Institute Investigator. The funders had no role in study design, data collection and analysis, decision to publish, or preparation of the manuscript.

**Competing interests:** C.H. is a scientific founder and a member of the scientific advisory board of Accent Therapeutics. Other authors have declared that no competing interests exist.

## Author summary

HIV-1 is known as a weak inducer of antiviral cytokines including IFN-I, but it is unclear how HIV-1 evades innate immunity. Different types of RNA modifications including m⁶A within the HIV-1 genome modulate viral replication; however, the role of m⁶A modifications of HIV-1 RNA in regulating innate immune responses remains elusive. Myeloid cells including macrophages are HIV-1 target cells and critical for generating antiviral immunity. In this study, we aimed to investigate the role of m⁶A modifications of HIV-1 RNA in regulating innate immune responses in myeloid cells. We found that m⁶A-modified HIV-1 RNA suppresses IFN-I expression in differentiated monocytic cells and primary macrophages. Our data suggest that the cellular protein RIG-I contributes to innate sensing of m⁶A-defective HIV-1 RNA in differentiated monocytic cells. Our findings provide new insights into the functions and mechanisms of m⁶A modifications of HIV-1 RNA in regulating innate immune sensing and responses in myeloid cells.

## Introduction

Transcriptional modification of RNA in cells plays a crucial role in its stability, transportation, processing and thus regulation of gene expression. There are more than 160 RNA modifications identified in eukaryotes [1]. Methylation at the $N^6$ position of adenosine (m⁶A) is a post-transcriptional RNA modification in internal and untranslated regions (UTRs) of eukaryotic mRNAs, microRNAs, small nuclear RNAs and long noncoding RNAs, which is important for RNA localization, stability and protein translation [1–5]. This methylation is controlled by two types of protein factors in cells, comprised of the writer complex [(methyltransferase-like 3 (METTL3) and METTL14] to incorporate methylation, and the erasers [fat mass and obesity associated protein (FTO) and α-ketoglutarate dependent dioxygenase AlkB homolog 5 (ALKBH5)] to remove m⁶A modification [6–9]. The RNA m⁶A modification is reversible and specifically recognized by a family of host proteins named m⁶A readers. The binding of m⁶A-modified RNA with the readers can significantly affect RNA trafficking, stability, localization, and translation [3, 8, 10].

RNA m⁶A modification has been discovered in several RNA and DNA viruses over the past 40 years, although its effects on the viral lifecycle remain not fully understood [11–16]. Increasing evidence suggests that m⁶A modification plays a major role in the regulation of viral replication and gene expression [17]. Recent advancements of RNA-sequencing based strategies expanded the identification and characterization of m⁶A to several clinically significant human pathogens [17], including HIV-1 [18–20]. Multiple m⁶A sites have been identified across the HIV-1 genomic RNA, although the m⁶A sites vary from different studies likely due to distinctive approaches and cell types used in these investigations [18–20]. By modulating protein expression of m⁶A writers, erasers and readers in HIV-1 producer and target cells, several studies demonstrated the importance of the m⁶A pathway in regulating HIV-1 replication and viral protein expression in CD4⁺ T cell lines and primary CD4⁺ T cells [18–22]. Moreover, HIV-1 infection of CD4⁺ T cells upregulates m⁶A levels of cellular RNA [23] and HIV-1 protease cleaves the antiviral m⁶A reader protein YTHDF3 in the viral particle [22]. Together, these studies indicate the complex interplay of the m⁶A pathway and host cells during HIV-1 infection.

The m⁶A pathway also plays an essential role in regulating the immune system [24]. In the early stage of virus infections, sensing viral nucleic acids in infected cells is a critical step to induce innate immune responses that can lead to the production of antiviral cytokines,

including IFN-I (mainly IFN-$\alpha$ and IFN-$\beta$) [25]. Genomic RNA of HIV-1 and other viruses can be detected by cytosolic sensors, including retinoic acid-induced gene I (RIG-I) and melanoma differentiation-associated gene 5 (MDA5) [25]. Detection of viral RNA by these sensors triggers activation of several cellular kinases, which phosphorylate interferon regulatory factors 3 and 7 (IRF3 and IRF7) to induce IFN-I expression [26, 27]. HIV-1 has been known as a weak inducer of host innate immune responses [28, 29], and it evades immune recognition by direct targeting of immune pathways, interacting with cellular proteins, or masking the viral genome from the cytosolic sensors [28, 30, 31]. HIV-1 RNA can be sensed by both RIG-I and MDA5, whereas it has evolved multiple strategies to escape innate immune surveillance [27, 32].

Myeloid cells, including the monocyte precursors and their differentiated macrophages and dendritic cells, are among the first targets during HIV-1 infection and transmission [32–34]. Myeloid cells are important for viral pathogen sensing and inflammation caused by HIV-1 infection [35, 36]. A recent study showed that 2′-*O*-methylation in HIV-1 RNA prevents MDA5-mediated sensing in monocytic cell lines, primary macrophages and dendritic cells, thereby reducing IFN-I induction and innate immune responses to HIV-1 [37]. However, the role of m6A in regulating innate immune responses to HIV-1 RNA and the underlying mechanisms have not been defined, which represents a significant knowledge gap in HIV-1 biology.

Here we show that m6A modifications of HIV-1 RNA reduce viral RNA sensing and the induction of IFN-I in differentiated monocytic cells and primary monocyte-derived macrophages (MDM). We observed that m6A-deficient HIV-1 RNA oligos induced higher IFN-I expression through the RIG-I-mediated pathway in differentiated monocytic cells, suggesting that m6A modification of viral RNA is an immune evasion strategy of HIV-1. These findings implicate that m6A modifications of HIV-1 RNA contribute to the regulation of innate immune responses to viral infection.

## Results

### A single m6A modification of HIV-1-derived RNA oligos inhibits IFN-I induction in U937 cells

To examine the effect of m6A modification of HIV-1 RNA on IFN-I induction, we designed two different RNA oligos corresponding to two fragments of HIV-1 genome each with or without a single m6A modification [21] for transfection experiments (Fig 1A). We have reported that these two m6A modifications in the 5′ untranslated regions (UTR) of HIV-1 genome are important for HIV-1 RNA binding to the m6A reader proteins (YTH domain family proteins 1–3) *in vitro* and viral replication in cells [21]. The m6A-modified RNA oligos 1 and 2 (both were 42 mer) contained a single m6A-modified adenosine in the conserved GG**A**CU motif of the HIV-1 (NL4-3 strain) genome [21]. We confirmed the m6A modification of these oligos by dot immunoblotting with equal amounts of RNAs using an m6A-specific antibody (Fig 1B and 1C). To mimic cellular responses to viral RNA in non-dividing macrophages, we differentiated monocytic U937 cells with phorbol 12-myristate 13-acetate (PMA) before transfection with the RNA oligos (Fig 1A). Compared to unmethylated control (Ctrl) RNA, m6A-modified RNA oligo 1 induced 3- to 4-fold lower ($P < 0.005$) levels of *IFN-α* and *IFN-β* mRNA in transfected U937 cells (Fig 1D). Similar results were obtained with transfection of oligo 2, although the effects were less significant compared to oligo 1 (Fig 1E). These results suggest that m6A modification of the 5′ UTR of HIV-1 RNA fragments can suppress IFN-I induction in differentiated U937 cells.

To examine whether the sequence of RNA oligos is important for m6A-mediated suppression of IFN-I expression in differentiated U937 cells, we designed a different pair of RNA oligo with or without a single m6A modification based on the scrambled sequence of HIV-1 RNA

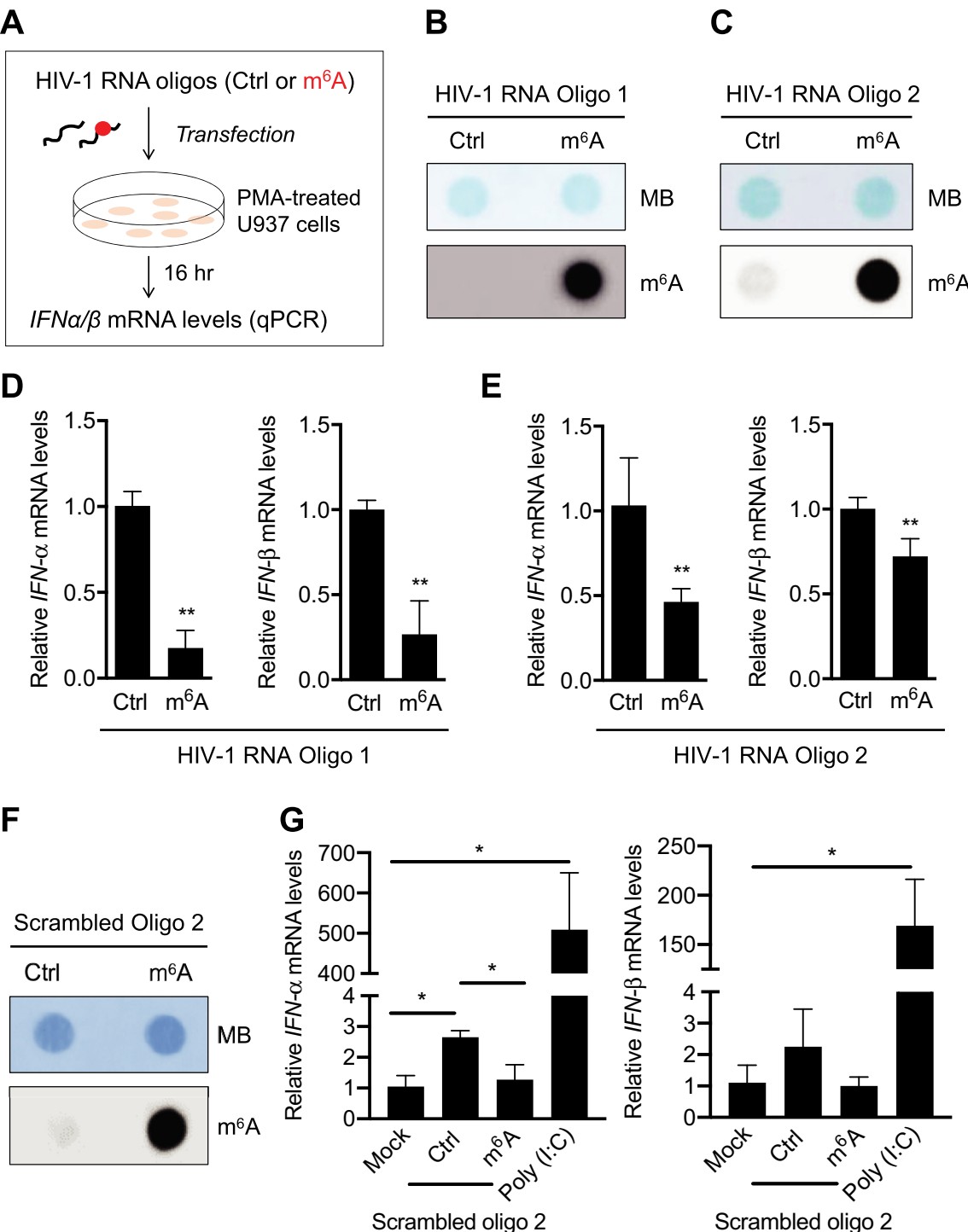

**Fig 1. A single m⁶A modification of HIV-1-derived RNA oligos inhibits IFN-I induction in differentiated U937 cells. (A)** A schematic of the experimental approach. **(B)** HIV-1 5′ UTR (nt. 235–281) RNA oligo 1 (50 ng), **(C)** HIV-1 5′ UTR (nt. 176–217) RNA oligo 2 (200 ng), or **(F)** sequence-scrambled oligo 2 (200 ng) with (m⁶A) or without (Ctrl) m⁶A modification were subjected to m⁶A dot-blot analysis. Methylene blue (MB) staining was used as an RNA loading control. **(D)** HIV-1 RNA oligo 1, **(E)** HIV-1 RNA oligo 2, or **(G)** sequence-scrambled oligo 2 (250 ng) with or without m⁶A modification were transfected into PMA-differentiated U937 cells (5×10⁵). At 16 hr post-transfection, *IFN-α* and *IFN-β* mRNA levels were measured by RT-qPCR. Data shown in D and E are means ± S.D. of three independent experiments. Data of panel G represent means ± S.D. of three biological replicates in one of two independent experiments. Un-paired t-test was used for statistical analysis. * *P* < 0.05, ** *P* < 0.005, compared with mock or Ctrl samples as indicated.

oligo 2. BLAST search of this 42-mer random RNA sequence through the NCBI nucleotide sequence database confirmed no significant similarity. We confirmed the m$^6$A modification of sequence-scrambled RNA oligo 2 by m$^6$A dot immunoblotting (Fig 1F). We then transfected PMA-differentiated U937 cells with these RNA oligos containing random sequences to measure *IFN-I* mRNA induction. Mock transfection was used as a background control and poly (I: C) transfection was used as a positive control. Compared to mock transfection, sequence-scrambled control oligo 2 (Ctrl) without m$^6$A modification induced 2-fold higher *IFN-I* mRNA expression, while m$^6$A-modified scrambled oligo 2 showed a background level (Fig 1G). Compared to the mock transfection, poly (I:C) transfection of U937 cells induced approximately 500- and 160-fold higher *IFN-α* and *IFN-β* mRNA levels ($P < 0.05$), respectively (Fig 1G). These results indicated that *IFN-I* mRNA expression induced by m$^6$A-deficient RNA oligos is likely independent of RNA sequence, suggesting a potentially general mechanism of m$^6$A modifications of viral RNA in evading innate immune responses to viral infection.

## Reduction of m$^6$A modifications of HIV-1 RNA by FTO increases IFN-I induction in U937 cells

The m$^6$A erasers (FTO and ALKBH5) orchestrate cellular mRNA functions by removing m$^6$A modifications on mRNA [2]. To investigate whether m$^6$A modifications of HIV-1 genomic RNA could suppress IFN-I induction in cells, we generated HIV-1 from proviral DNA-transfected HEK293T cells and treated isolated viral RNA with recombinant FTO to reduce m$^6$A level before RNA transfection into PMA-differentiated U937 cells (Fig 2A). Isolated RNA from purified HIV-1 virions was demethylated with recombinant FTO *in vitro*, resulting in a 10-fold decrease in m$^6$A level relative to control HIV-1 RNA (Fig 2B). Transfection of m$^6$A-reduced HIV-1 RNA into PMA-differentiated U937 cells induced 3-fold higher *IFN-α* and *IFN-β* expression ($P < 0.0005$) compared to control HIV-1 RNA (Fig 2C), suggesting that m$^6$A modification of HIV-1 genomic RNA suppresses IFN-I induction in myeloid cells.

## Infection of U937 cells with m$^6$A-deficient HIV-1 increases IFN-I expression and phosphorylation of IRF3 and IRF7

To determine the effect of m$^6$A of HIV-1 RNA on IFN-I induction during viral infection, HIV-1 containing lower levels of m$^6$A in viral RNA was generated by overexpression of the eraser FTO in HEK293T cells and was used to infect PMA-differentiated U937 cells (Fig 3A). Compared to the vector control, FTO overexpression in HEK293T cells had no significant effect on the expression of HIV-1 Gag and capsid (CA, or p24) proteins (Fig 3B). HIV-1 derived from FTO-overexpressed HEK293T cells (m$^6$A-deficient HIV-1) showed 10-fold lower m$^6$A levels of viral RNA compared to viruses derived from control cells (Fig 3C). To examine the infectivity of m$^6$A-deficient HIV-1 relative to the control virus, we infected TZM-bl cells with HIV-1 using three different amounts of p24 as viral input and detected the infectivity at 48 hpi. TZM-bl cells are widely used as HIV-1 indicator cells because they express luciferase upon productive HIV-1 infection [38, 39]. The infectivity of m$^6$A-deficient HIV-1 and control virus was comparable with the same amount of p24 (Fig 3D), indicating that FTO overexpression in HIV-1 producer cells does not alter viral infectivity.

At 16 hr post-infection (hpi), PMA-differentiated U937 cells infected with m$^6$A-deficient HIV-1 expressed around 2-fold higher *IFN-α* and *IFN-β* mRNA ($P < 0.05$) relative to cells infected with control HIV-1 (Fig 3E). During the early stage of viral infection, IFN-I expression is predominately driven by IRF3 and IRF7 after their activation through phosphorylation [25, 40]. We thus tested whether m$^6$A-deficient HIV-1 affected phosphorylation of IRF3 and IRF7. Compared to mock infection, U937 cells infected with control HIV-1 increased the levels

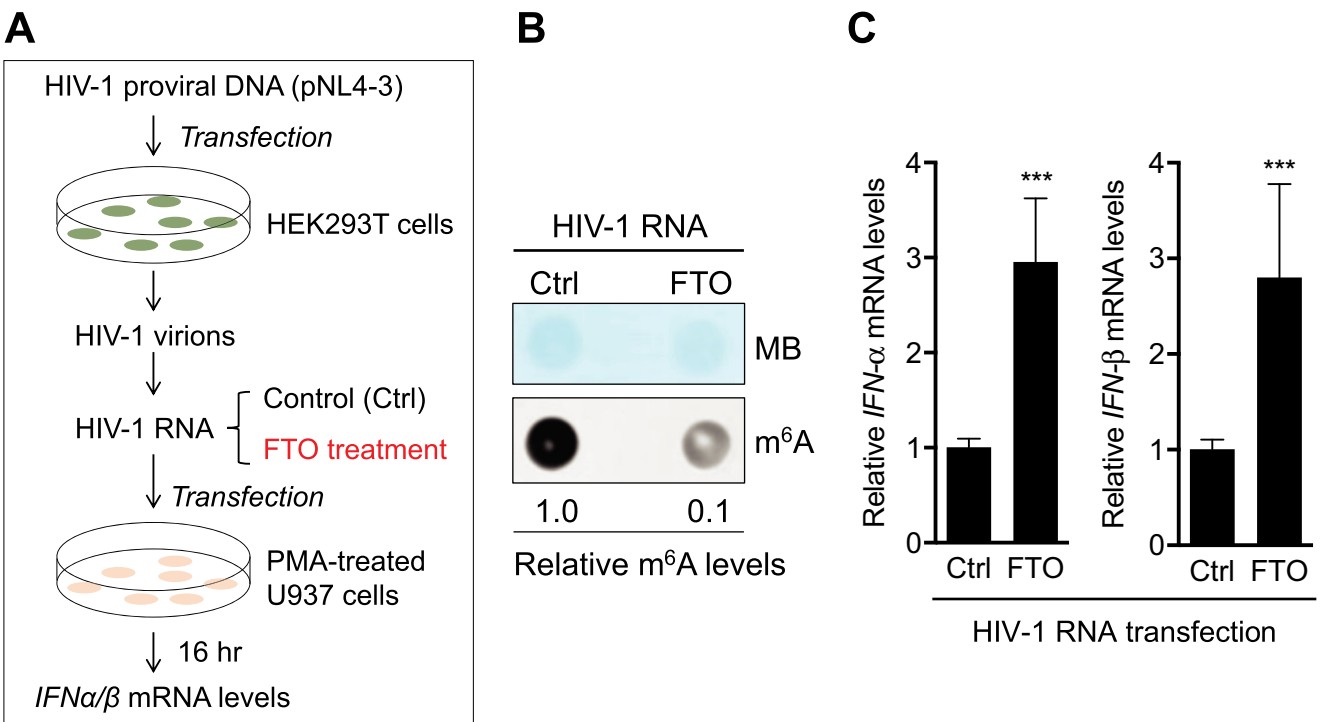

**Fig 2. Inhibition of m⁶A modifications of HIV-1 RNA by recombinant FTO increases IFN-I induction. (A)** A schematic of the experimental approach. **(B)** m⁶A levels of HIV-1 genomic RNA were reduced by treatment with recombinant demethylase FTO and 50 ng of RNAs were used to confirm the m⁶A levels by dot-blotting. Methylene blue (MB) staining was used as an RNA loading control. **(B)** and **(C)** 250 ng of the above RNAs were transfected in PMA-differentiated U937 cells. At 16 hr post-transfection, *IFN-α* and *IFN-β* mRNA levels were measured by RT-qPCR. The results are shown as means ± S.D. of three independent experiments. Mann-Whitney t-test was used for statistical analysis. *** $P < 0.0005$, compared with control (Ctrl) samples.

of phosphorylation of IRF3 and IRF7 1.6- and 1.8-fold at 4 hpi, respectively. Of note, the levels of phosphorylation of IRF3 and IRF7 in U937 cells infected with m⁶A-deficient HIV-1 were 2.6- and 3.1-fold higher than those in mock infection, and 1.6- to 1.7-fold higher compared to cells infected with control HIV-1 (Fig 3F). Together, these results suggest that m⁶A modifications of HIV-1 RNA evade IFN-I induction and IRF3/7 activation during the early stage of HIV-1 infection in differentiated monocytic cells.

## Blocking HIV-1 reverse transcription partially reduces m⁶A-deficient HIV-1 induced IFN-I expression in U937 cells

To investigate whether HIV-1 reverse transcription and viral cDNA are important for IFN-I induction by m⁶A-deficient HIV-1, we treated PMA-differentiated U937 cells with the reverse transcriptase (RT) inhibitor nevirapine (NVP) to block HIV-1 reverse transcription and productive viral infection (Fig 4A). In the absence of NVP, the level of *gag* mRNA in U937 cells infected with m⁶A-deficient HIV-1 (FTO) was 28-fold higher compared to cells infected with control HIV-1 (Vector) (Fig 4B). NVP treatment efficiently reduced HIV-1 *gag* mRNA expression 3- to 8-fold ($P \leq 0.0001$) in U937 cells infected with control HIV-1 or m⁶A-deficient HIV-1 at 16 hpi, respectively (Fig 4B). Compared to U937 cells infected with control HIV-1, cells infected with m⁶A-deficient HIV-1 induced 4- and 3-fold higher *IFN-α* and *IFN-β* mRNA expression at 16 hpi, respectively (Fig 4C). Moreover, NVP treatment partially reduced the levels of *IFN-α* and *IFN-β* mRNA in U937 cells infected with control or m⁶A-deficient HIV-1.

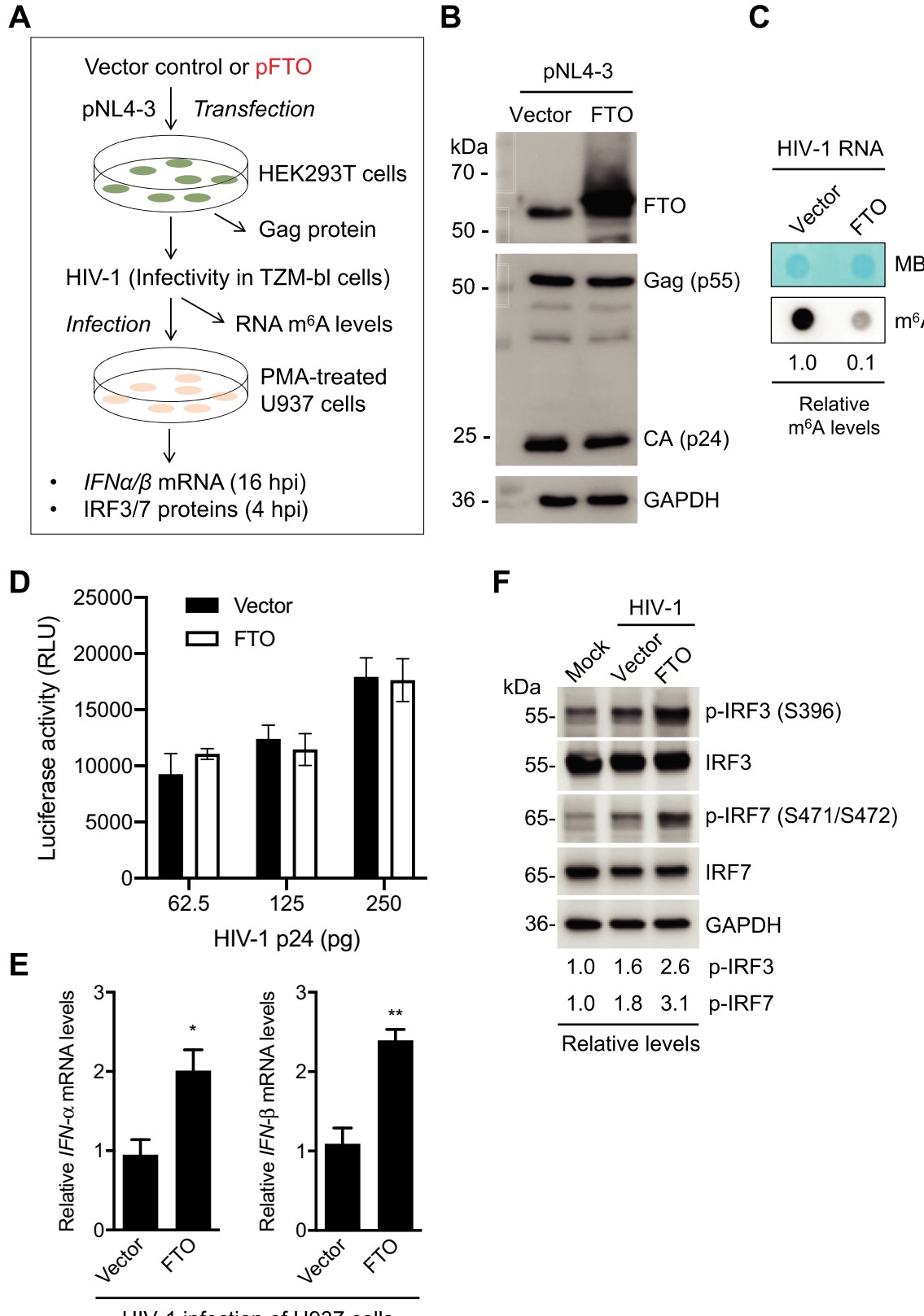

**Fig 3. Infection of U937 cells with m⁶A-deficient HIV-1 increases *IFN-I* expression and IRF3/7 phosphorylation. (A)** A schematic diagram of the experimental approach. **(B)** HEK293T cells were transfected with vector control or an FTO-expressing plasmid (FTO). After 24 hr, HIV-1 proviral DNA clone (pNL4-3) was transfected for 48 hr. Then, cell lysates were collected, and Western blotting was performed using indicated antibodies. **(C)** the m⁶A levels in HIV-1 genomic RNA were determined by the dot-blot assay using 100 ng purified viral RNA derived from vector control or FTO-expressing HEK293T cells. Methylene blue (MB) staining was used as an RNA loading control. **(D)** The HIV-1 indicator TZM-bl cells were infected with the HIV-1 derived from vector control or FTO-overexpressing HEK293T cells. Medium was used as a mock control of the infection. TZM-bl cells were lysed 48 hpi for the detection of luciferase activity. All luciferase values were normalized to 10 μg protein. RLU, relative light units. **(E)** PMA-differentiated U937 cells were infected with HIV-1 (250 pg of p24) generated from vector control or FTO-expressing HEK293T cells for 16 hr, and *IFN-α* and *IFN-β* mRNA levels were quantified by RT-qPCR. The results are shown as means ± S.D. of three independent experiments. Un-paired t-test was used for statistical analysis. * $P < 0.05$, ** $P < 0.005$, vector controls were normalized with mock infection samples. **(F)** Mock or HIV-1 (250 pg of p24) infected U937 cell lysates were collected at 4 hr post-infection (hpi) for Western blotting of IRF3 and IRF7 expression. GAPDH is a loading control. Relative levels of phosphorylated IRF3 (p-IRF3 at S396) and phosphorylated IRF7 (p-IRF7 at S471/S472) are shown.

With NVP treatment, *IFN-I* mRNA levels in U937 cells infected with m⁶A-deficient HIV-1 were 2- to 3-fold higher relative to cells infected with control HIV-1 (Fig 4C). These results suggest that m⁶A-deficient HIV-1 genomic RNA triggers IFN-I induction in differentiated monocytic cells, at least in part.

## Reduction of m⁶A modifications of HIV-1 RNA by ALKBH5 increases IFN-I induction

To confirm the results of FTO treatment and overexpression, we also examined the effect of another m⁶A eraser ALKBH5 on HIV-1 RNA-mediated IFN-I induction in PMA-differentiated U937 cells (Fig 5A). ALKBH5 overexpression in HIV-1-producing HEK293T cells had no significant effect on the expression of HIV-1 Gag and CA (Fig 5B). The m⁶A modification of HIV-1 RNA generated from ALKBH5-overexpressed HEK293T cells showed a 2-fold decrease compared to HIV-1 RNA from control cells (Fig 5C). *IFN-α* and *IFN-β* levels in U937 cells transfected with HIV-1 RNA from ALKBH5-overexpressed HEK293T cells were 1.8-fold higher ($P < 0.05$) compared to that from control cells (Fig 5D). Infection of U937 cells with HIV-1 from ALKBH5-overexpressed HEK293T cells induced 2-fold higher *IFN-α* and *IFN-β* expression ($P < 0.0005$) compared to HIV-1 from control HEK293T cells (Fig 5E). Furthermore, infection of U937 cells with m⁶A-deficient HIV-1 generated from ALKBH5-overexpressed HEK293T cells induced approximately 2-fold higher phosphorylation of IRF3 and IRF7 when compared with control HIV-1 (Fig 5F). Thus, inhibition of m⁶A modifications of HIV-1 RNA by eraser overexpression in virus-producing cells increases *IFN-I* induction and activation of IRF3 and IRF7 in differentiated U937 cells.

## Pharmacological inhibition of m⁶A modification of HIV-1 RNA induces IFN-I expression

We next investigated pharmacological inhibition of m⁶A modification of HIV-1 RNA using 3-deazaadenosine (DAA), an inhibitor of S-Adenosylhomocysteine (SAH) hydrolase that can catalyze the reversible hydrolysis of SAH to adenosine and homocysteine [41]. DAA causes SAH accumulation thereby elevating the ratio of SAH to S-adenosylmethionine (SAM), a substrate of m⁶A modification, and subsequent inhibition of SAM-dependent methyltransferases [41]. We generated HIV-1 from HEK293T cells and performed HIV-1 RNA transfection and viral infection to evaluate IFN-I induction in PMA-differentiated U937 cells (Fig 6A). DAA-treatment of HEK293T cells did not affect HIV-1 production and release (Fig 6B), but reduced m⁶A level in HIV-1 RNA 7-fold compared to control cells (Fig 6C). Transfection of PMA-differentiated U937 cells with purified RNA from HIV-1 produced from DAA-treated HEK293T cells (DAA-HIV-1) induced 15-fold and 2.3-fold higher *IFN-α* and *IFN-β* expression

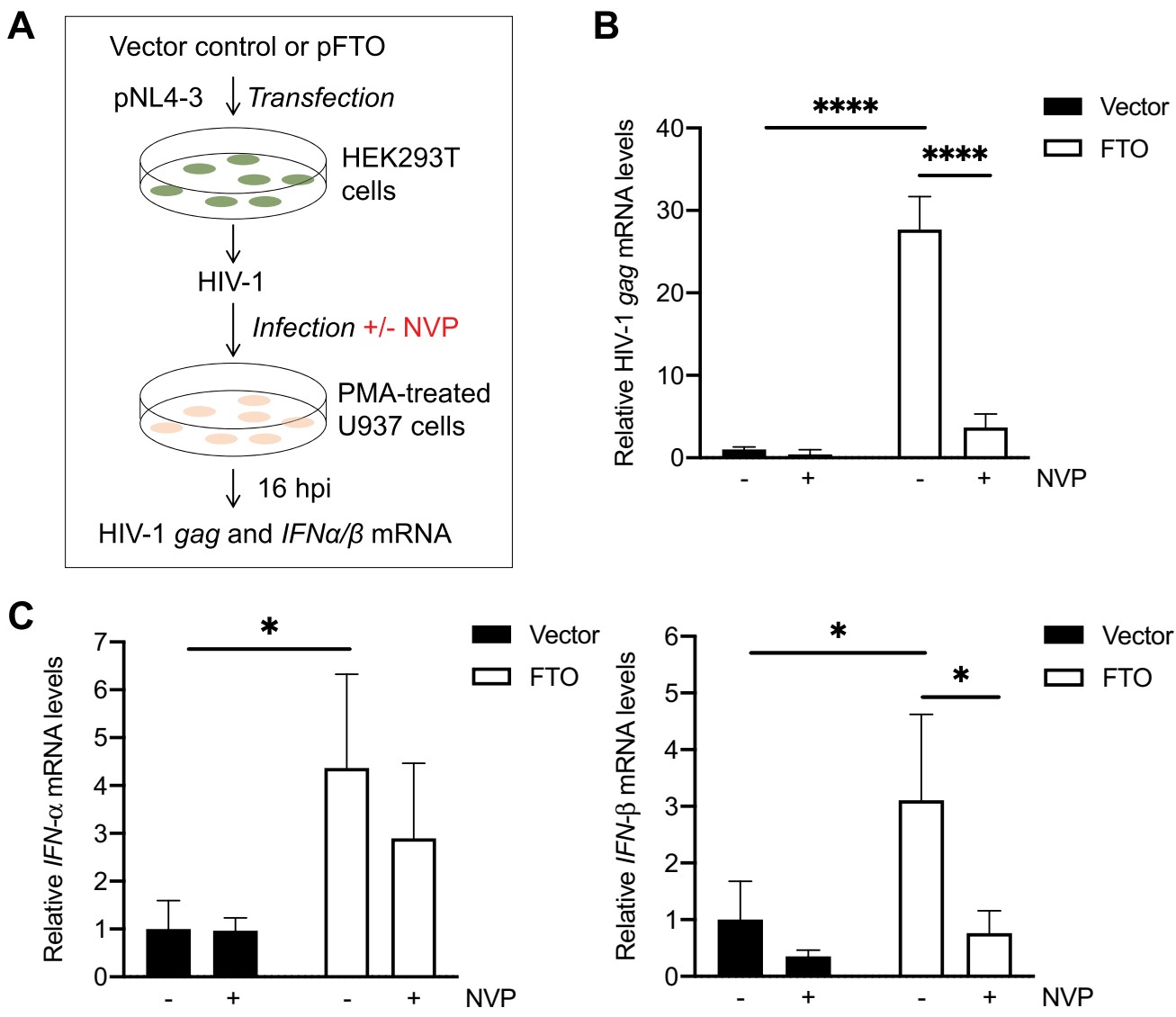

**Fig 4. Blocking HIV-1 reverse transcription partially reduces m⁶A-deficient HIV-1 induced *IFN-I* expression in U937 cells. (A)** A schematic diagram of the experimental approach. **(B-C)** PMA-differentiated U937 cells in the nevirapine (NVP) groups were pre-treated with NVP (5 μM) for 2 hr prior to 2 hr incubation with HIV-1$_{NL4-3}$ (250 pg of p24) generated from vector control or FTO-overexpressing HEK293T cells. NVP (5 μM) was maintained in the medium throughout the infection and subsequent culture. At 16 hpi, mRNA levels of **(B)** HIV-1 *gag* and **(C)** *IFN-α* and *IFN-β* were quantified by RT-qPCR. Spliced *GAPDH* was used as a normalization control (B-C). The results are shown as means ± S.D. of triplicated samples. One representative experiment of four repeats is shown. Two-way ANOVA was used for statistical analysis. * $P \leq 0.05$, **** $P \leq 0.0001$.

($P < 0.0005$), respectively (Fig 6D). Moreover, infection of PMA-differentiated U937 cells with DAA-HIV-1 induced a 2-3-fold increase in *IFN-I* expression ($P < 0.0005$) compared to viruses from control HEK293T cells (Fig 6E). These data further validate that m⁶A of HIV-1 RNA suppresses IFN-I induction in differentiated monocytic cells.

## Knockout (KO) of erasers increases m⁶A levels in HIV-1 RNA and reduces IFN-I induction

To validate the results from eraser overexpression, we constructed FTO-KO and ALKBH5-KO HEK293T cell lines by the CRISPR-Cas9 method. Next, these cell lines were transfected to

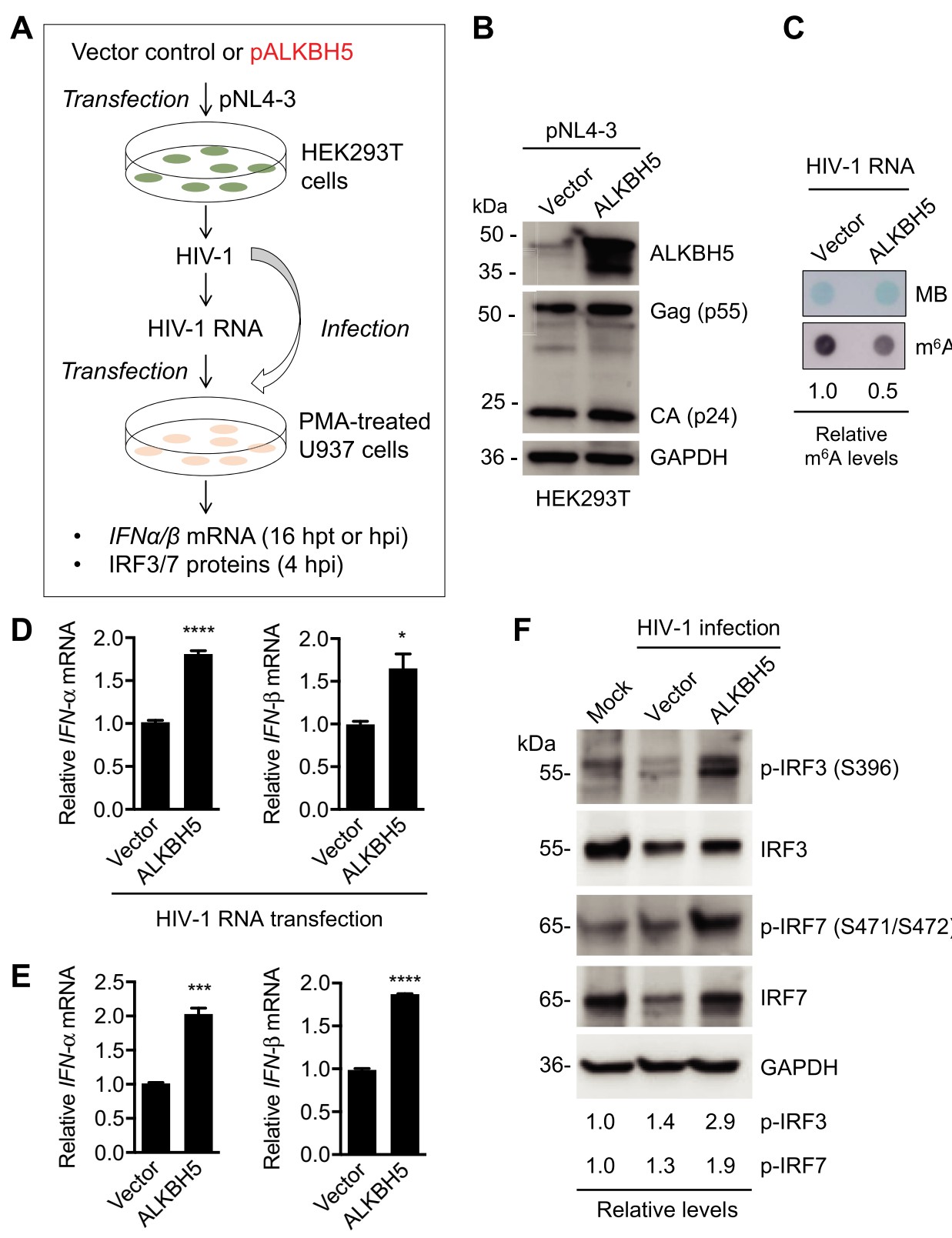

**Fig 5. Inhibition of m⁶A modifications of HIV-1 RNA by ALKBH5 increases IFN-I induction. (A)** A schematic diagram of the experimental approach. **(B)** HEK293T cells were transfected with a vector control or an ALKBH5-expressing plasmid (ALKBH5). After 24 hr, pNL4-3 was transfected into these cells for 48 hr. Western blotting of cell lysates was performed using specific antibodies. **(C)** HIV-1 genomic RNA m⁶A levels were determined by the dot-blot assay using 100 ng viral RNA from vector control or ALKBH5-expressing HEK293T cells. Methylene blue (MB) staining was used as an RNA loading control. **(D)** PMA-differentiated U937 cells were transfected with 500 ng of the indicated HIV-1 RNAs. At 16 hr post-transfection (hpt), cells were collected for the analysis of *IFN-α* and *IFN-β* mRNA levels by RT-qPCR. The results are shown as means ± S.D. of three repeated assays. * $P < 0.05$, **** $P < 0.0001$. **(E)** PMA-differentiated U937 cells were infected with HIV-1 (250 pg of p24) from vector control or ALKBH5-expressing HEK293T cells for 16 hr, and *IFN-α* and *IFN-β* mRNA levels were quantified by RT-qPCR. The results are shown as means ± S.D. of three repeated experiments. Vector controls were normalized with non-infection samples. Un-paired t-test was used for statistical analysis. *** $P < 0.0005$, **** $P < 0.0001$. **(F)** PMA-differentiated U937 cells were infected with HIV-1 (250 pg of p24) from vector control or ALKBH5-overexpressing HEK293T cells. At 4 hpi, cell lysates were analyzed by Western blot with the indicated antibodies. GAPDH was used as a loading control. Relative levels of phosphorylated IRF3 (pIRF3 at S396) and phosphorylated IRF7 (pIRF7 at S471/S472) are shown.

generate HIV-1 with increased m⁶A of viral RNA for transfection or infection assays (Fig 7A). Western blotting results showed that FTO and ALKBH5 were completely silenced and HIV-1 Gag protein expression was not significantly affected by FTO and ALKBH5 knockout (Fig 7B). HIV-1 RNA from FTO-KO and ALKBH5-KO cells showed 7- and 25-fold higher m⁶A levels, respectively, relative to that from control (Ctrl-KO) cells (Fig 7C). Transfection of PMA-differentiated U937 cells with HIV-1 RNA derived from FTO-KO or ALKBH5-KO cells showed a 3-4-fold decrease ($P < 0.05$) in *IFN-I* expression compared to that from Ctrl-KO cells (Fig 7D). Moreover, infection of PMA-differentiated U937 cells with HIV-1 from FTO-KO or ALKBH5-KO cells induced approximately 2-fold less *IFN-I* expression ($P < 0.005$) compared to Ctrl-KO cells (Fig 7E). Thus, increasing m⁶A levels in HIV-1 RNA by eraser KO in virus-producing cells reduces *IFN-I* induction in differentiated monocytic cells.

## *IFN-I* mRNA expression is rescued in U937 cells transfected with m⁶A-increased HIV-1 RNA upon recombinant FTO treatment

To further validate the above results, we performed a rescue experiment by treating the HIV-1 RNA from FTO-KO HEK293T cells with recombinant FTO *in vitro* and then transfecting the derived HIV-1 RNA into PMA-differentiated U937 cells (Fig 8A). Recombinant FTO treatment of HIV-1 RNA from FTO-KO cells reduced m⁶A level approximately 3-fold, while the level was still 5.5-fold higher than that of control HIV-1 (Fig 8B). Consistently, we observed reduced IFN-I induction in U937 cells transfected with HIV-1 RNA derived from FTO-KO cells compared to control HIV-1 RNA (Fig 8C). Recombinant FTO-treated HIV-1 RNA from FTO-KO cells demonstrated rescue effects on *IFN-I* mRNA induction in U937 cells compared to HIV-1 RNA from FTO-KO cells (Fig 8C). These results further support the notion that m⁶A-modified HIV-1 RNA reduces innate immune sensing and IFN-I induction in monocytic cells.

## Transfection of m⁶A-altered HIV-1 RNA into primary MDM modulates IFN-I expression

To validate the results obtained from PMA-differentiated U937 cells, we generated MDM with CD14⁺ monocytes isolated from de-identified healthy blood donors [42] and examined the effect of m⁶A-decreased or m⁶A-increased HIV-1 RNA on IFN-I expression in MDM. In this experiment, m⁶A levels of HIV-1 RNA in virions produced from FTO-overexpressing (FTO-OE) HEK293T cells were 2-fold lower compared to that from vector control cells (Fig 9A, left panel). In contrast, m⁶A levels of HIV-1 RNA in virions produced from FTO-KO HEK293T cells were 14-fold higher compared to that from control KO (Ctrl-KO) cells (Fig 9A, right panel). IFN-I expression was undetectable in mock transfection (no RNA) of MDM as a background control (Fig 9B–9E). Transfection of m⁶A-decreased HIV-1 RNA (FTO-OE) into MDM induced 22- to 34-fold higher *IFN-I* mRNA expression ($P < 0.05$) compared to control HIV-1

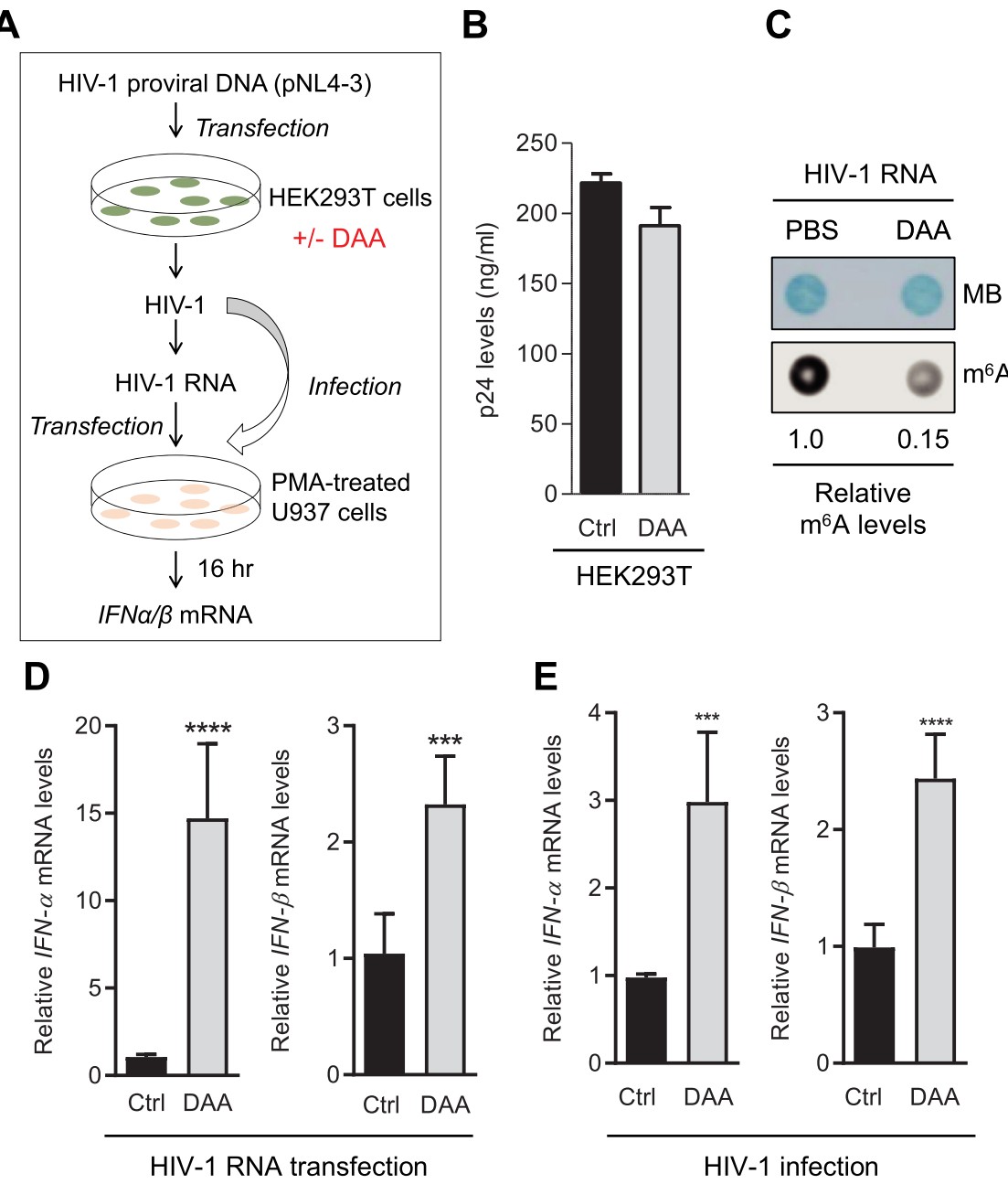

**Fig 6. DAA-treatment reduces m⁶A modifications of HIV-1 RNA and increases IFN-I induction.** (A) A schematic diagram of the experimental approach. HEK293T cells were treated with the solvent PBS control (Ctrl) or DAA (50 μM) for 3 hr and then transfected with the HIV-1 proviral DNA pNL4-3. HIV-1 in the supernatants was collected 48 hr later. (B) HIV-1 p24 levels in the supernatants of HEK293T cells were measured by ELISA. (C) RNA (100 ng) from these viruses used for the m⁶A dot-blot assay. Methylene blue (MB) staining was used as an RNA loading control. (D) HIV-1 RNA (250 ng) from Ctrl and DAA-treated samples were transfected into PMA-differentiated U937 cells. After 16 hr, cells were collected for the analysis of *IFN-α* and *IFN-β* mRNA levels by RT-qPCR. The results are shown as means ± S.D. of three independent experiments. *** $P < 0.0005$, **** $P < 0.0001$. Un-paired t-test was used for statistical analysis. (E) HIV-1 (250 pg of p24) from HEK293T cells was used to infect PMA-differentiated U937 cells. At 16 hpi, U937 cells were collected for the analysis of *IFN-I* mRNA levels by RT-qPCR. The results are shown as means ± S.D. of three independent experiments. *** $P < 0.0005$, **** $P < 0.0001$, Ctrl samples were normalized with mock infection samples. Un-paired t-test was used for statistical analysis.

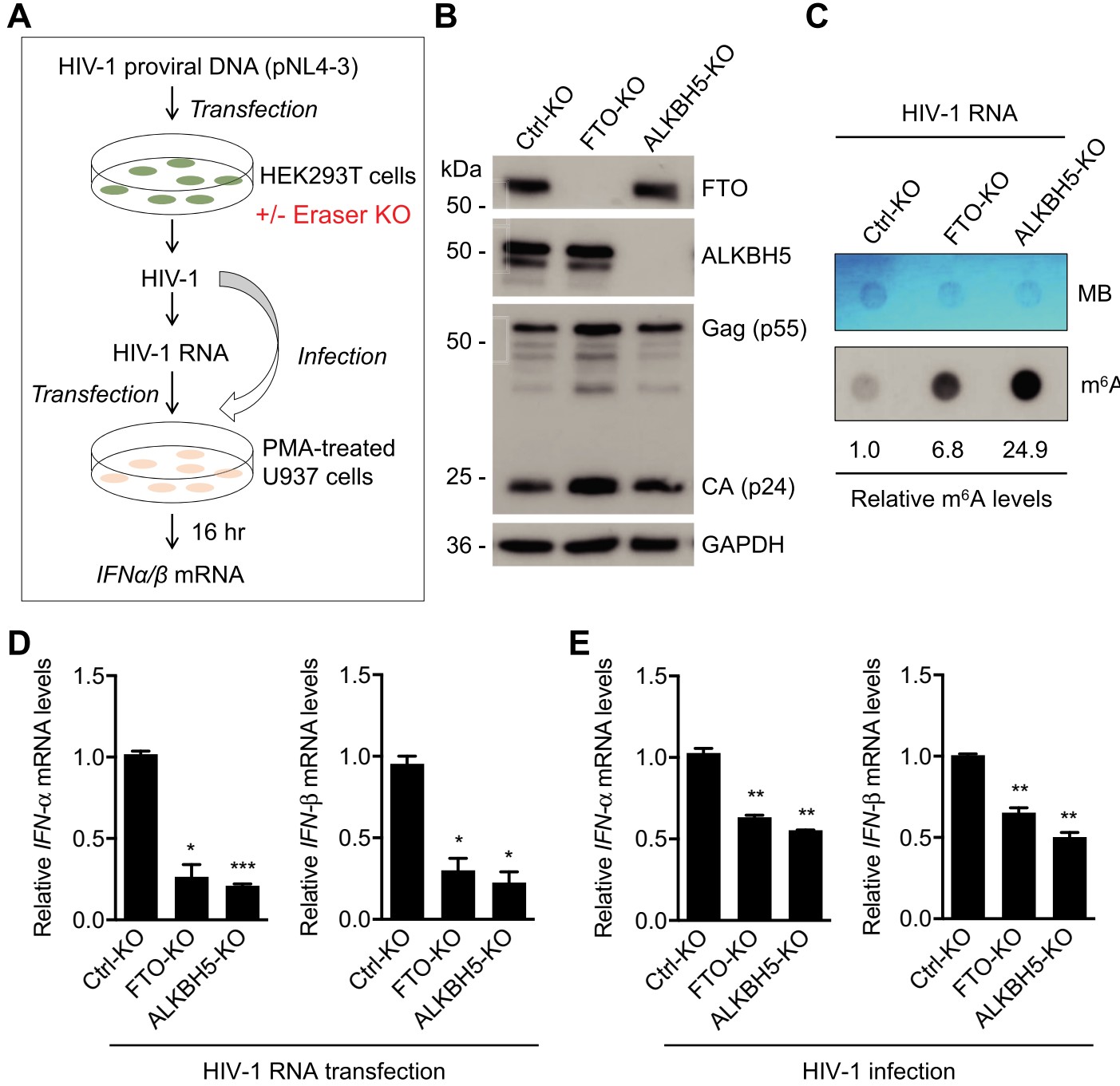

**Fig 7. Knockout of erasers increases m⁶A levels in HIV-1 RNA and reduces IFN-I induction. (A)** A schematic diagram of the experimental approach. **(B)** A single clone-derived control, FTO or ALKBH5 knockout (KO) HEK293T cells were transfected with the HIV-1 proviral DNA pNL4-3. After 48 hr, cells were collected for Western blotting analysis. **(C)** HIV-1 from the KO cells were collected and viral genomic RNA m⁶A level was determined by the dot-blot assay using 200 ng viral RNA. Methylene blue (MB) staining was used as an RNA loading control. **(D)** HIV-1 RNA (250 ng) from KO cells were transfected into PMA-differentiated U937 cells. After 16 hr, cells were collected for the analysis of IFN-α and IFN-β mRNA levels by RT-qPCR. The results are shown as means ± S.D. of three repeats with similar result. * $P < 0.05$, *** $P < 0.0005$. Un-paired t-test was used for statistical analysis. **(E)** HIV-1 (250 pg of p24) from KO cells were used to infect PMA-differentiated U937 cells for 16 hr, and cells were collected for the analysis of IFN-α and IFN-β mRNA levels by RT-qPCR. The results are shown as means ± S.D. of three repeats with similar result. ** $P < 0.005$. Un-paired t-test was used for statistical analysis.

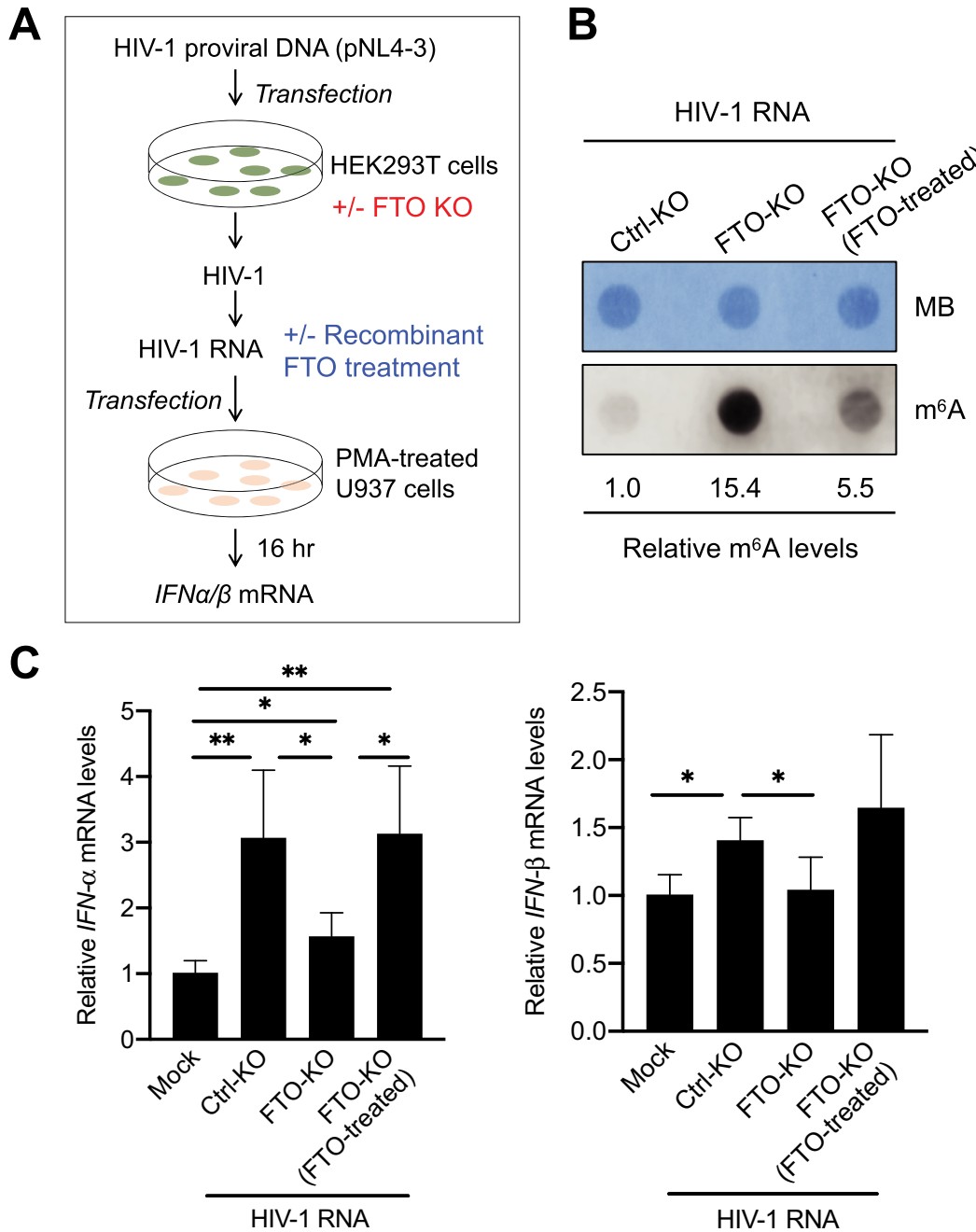

**Fig 8. *IFN-I* mRNA expression is rescued in U937 cells transfected with m⁶A-increased HIV-1 RNA upon recombinant FTO treatment. (A)** A schematic diagram of the experimental approach. **(B)** HIV-1 RNA isolated from virions generated from FTO-KO HEK293T cells was treated with or without recombinant FTO. HIV-1 RNA derived from Ctrl-KO HEK293T cells was used as a control. m⁶A levels of HIV-1 RNA (50 ng/sample) were measured by the m⁶A dot-blot assay. Methylene blue (MB) staining was used as an RNA loading control. **(C)** HIV-1 RNA (250 ng) was transfected in PMA-differentiated U937 cells ($5 \times 10^5$). Mock transfection of U937 cells without RNA was used as a negative control. At 16 hr post-transfection, *IFN-α* and *IFN-β* mRNA levels were measured by RT-qPCR. The results are shown as means ± S.D. of four biological samples from two independent experiments. * $P < 0.05$, ** $P < 0.01$. Un-paired t-test was used for statistical analysis.

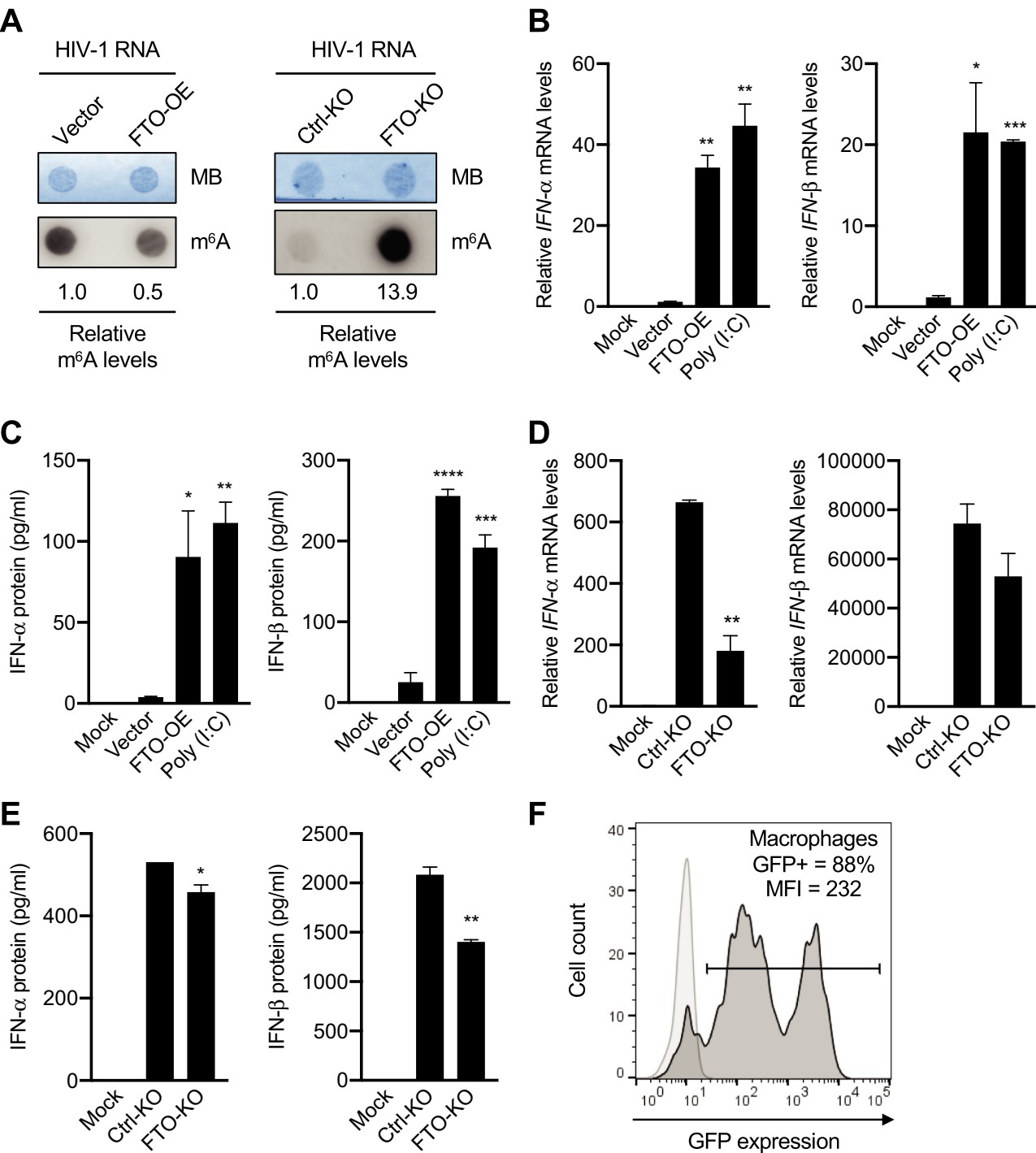

**Fig 9. Transfection of m⁶A-altered HIV-1 RNA into primary MDM modulates IFN-I expression. (A)** m⁶A levels of HIV-1 RNA (100 ng) of virions generated from vector control HEK293T cells, FTO-overexpressed HEK293T cells (FTO-OE), control-KO HEK293T cells (Ctrl-KO), or FTO-KO HEK293T cells were measured by dot-blotting. Methylene blue (MB) staining was used as an RNA loading control. **(B-E)** MDM (5×10⁵) were mock-transfected (no RNA) or transfected with 125 ng isolated HIV-1 RNA or poly (I:C) as indicated. At 24 hr post-transfection, **(B and D)** *IFN-I* mRNA levels in MDM were measured by qRT-PCR. **(C and E)** IFN-I protein levels in the supernatants of transfected MDM were detected by ELISA. These experiments were repeated three times with MDM from three different donors. Data shown were from one representative experiment with means ± S.D. of three biological repeats. Un-paired t-test was used for statistical analysis. * $P<0.05$, ** $P<0.01$, *** $P<0.001$, **** $P<0.0001$ compared to the indicated controls (B-C, compared to the vector control; D-E, compared to the Ctrl-KO). (B-E) IFN-I levels in mock samples were undetectable and used as a background control. **(F)** Flow cytometry detection of GFP expression in MDM mock-transfected (light grey peak) or transfected with *gfp* mRNA (dark grey peaks). Percentage of GFP-positive cells (88%) and mean fluorescence intensity (MFI) are shown.

RNA (Vector) (Fig 9B). IFN-I protein levels in the supernatants of MDM transfected with m$^6$A-decreased HIV-1 RNA were 10- to 24-fold higher ($P$ <0.01) compared to control HIV-1 RNA (Fig 9C). As positive controls, poly (I:C) transfection of MDM induced significantly higher IFN-I mRNA and protein expression ($P$ <0.01) compared to the control HIV-1 RNA (Fig 9B and 9C). Furthermore, transfection of m$^6$A-increased HIV-1 RNA (FTO-KO) into MDM induced 3.7-fold lower *IFN-α* ($P$ < 0.001) and 1.4-fold lower *IFN-β* mRNA expression compared to control HIV-1 RNA (Ctrl-KO) (Fig 9D). IFN-I protein levels in the supernatants of MDM transfected with m$^6$A-increased HIV-1 RNA were significantly lower ($P$ <0.05) compared to control HIV-1 RNA (Fig 9E). The transfection efficiency of MDM was around 86% based on a GFP mRNA control (Fig 9F). Using MDM from three different healthy donors, we obtained similar results with donor variations among experiments. Together, these results demonstrated that m$^6$A-modified HIV-1 RNA evades innate immune sensing in primary MDM.

## RIG-I, but not MDA5, is important for sensing HIV-1 RNA lacking m$^6$A modification

To characterize the cellular sensing mechanisms of m$^6$A-deficient HIV-1 RNA, RIG-I and MDA5 in U937 cells were silenced by KO and shRNA, respectively. RIG-I-KO U937 cells were constructed and undetectable RIG-I expression was confirmed (Fig 10A). To test whether these cells responded to RNA stimulation, poly(I:C) was transfected into cells and *IFN-I* expression was measured. Compared to mock-transfected cells, poly(I:C) transfection induced high levels of *IFN-I* expression in RIG-I-KO and control U937 cells (Fig 10B). As expected, the induction of *IFN-I* by poly(I:C) was significantly reduced by 2-fold in RIG-I-KO U937 cells ($P$ < 0.005) compared to control cells (Fig 10B), confirming that RIG-I acted as an RNA sensor to induce *IFN-I* expression in these cells. In control U937 cells, transfection of single m$^6$A-modified HIV-1 RNA oligos induced lower *IFN-I* expression ($P$ < 0.0001) compared to unmethylated RNA oligo counterparts (Fig 10C and 10D). However, in RIG-I-silenced U937 cells, transfection of m$^6$A-modified HIV-1 RNA oligos had no effect on *IFN-I* expression relative to unmethylated control oligos (Fig 10C and 10D), suggesting a pivotal role of RIG-I in sensing m$^6$A-deficient HIV-1 RNA.

Furthermore, we examined the potential role of MDA5 in sensing m$^6$A-deficient HIV-1 RNA in monocytic cells. MDA5 expression was substantially reduced by 97% in differentiated U937 cells with MDA5 knockdown (shMDA5) compared to vector control (shCtrl) cells (Fig 11A). As a positive control, poly(I:C) transfection induced high levels of *IFN-I* expression in both shCtrl and shMDA5 U937 cells. As expected, poly(I:C) transfection into shMDA5 U937 cells significantly decreased *IFN-I* levels relative to shCtrl cells (Fig 11B). These cells were then examined for their ability to induce *IFN-I* expression by HIV-1 5′ UTR RNA oligos with or without single m$^6$A modification [21]. Compared to unmethylated HIV-1 RNA oligos, transfection of m$^6$A-modified HIV-1 RNA oligos reduced *IFN-I* expression in both shCtrl and shMDA5 U937 cells (Fig 11C and 11D), suggesting that MDA5 is not a specific cellular sensor to detect m$^6$A-deficient HIV-1 RNA in differentiated monocytic cells.

## Discussion

HIV-1 genomic RNA contains 10–14 sites of m$^6$A modifications in the 5′-, 3′-UTR and several coding regions [18–20]. Recent studies indicate that m$^6$A modification has important effects on HIV-1 replication, gene expression, and host responses to viral infection [17, 21–23]. It has been shown that cellular enzymes involved in RNA m$^6$A modifications negatively regulate the innate immune response to infection of human cytomegalovirus, influenza A virus, adenovirus, or vesicular stomatitis virus by targeting the IFN-I pathway [43, 44]. A recent study

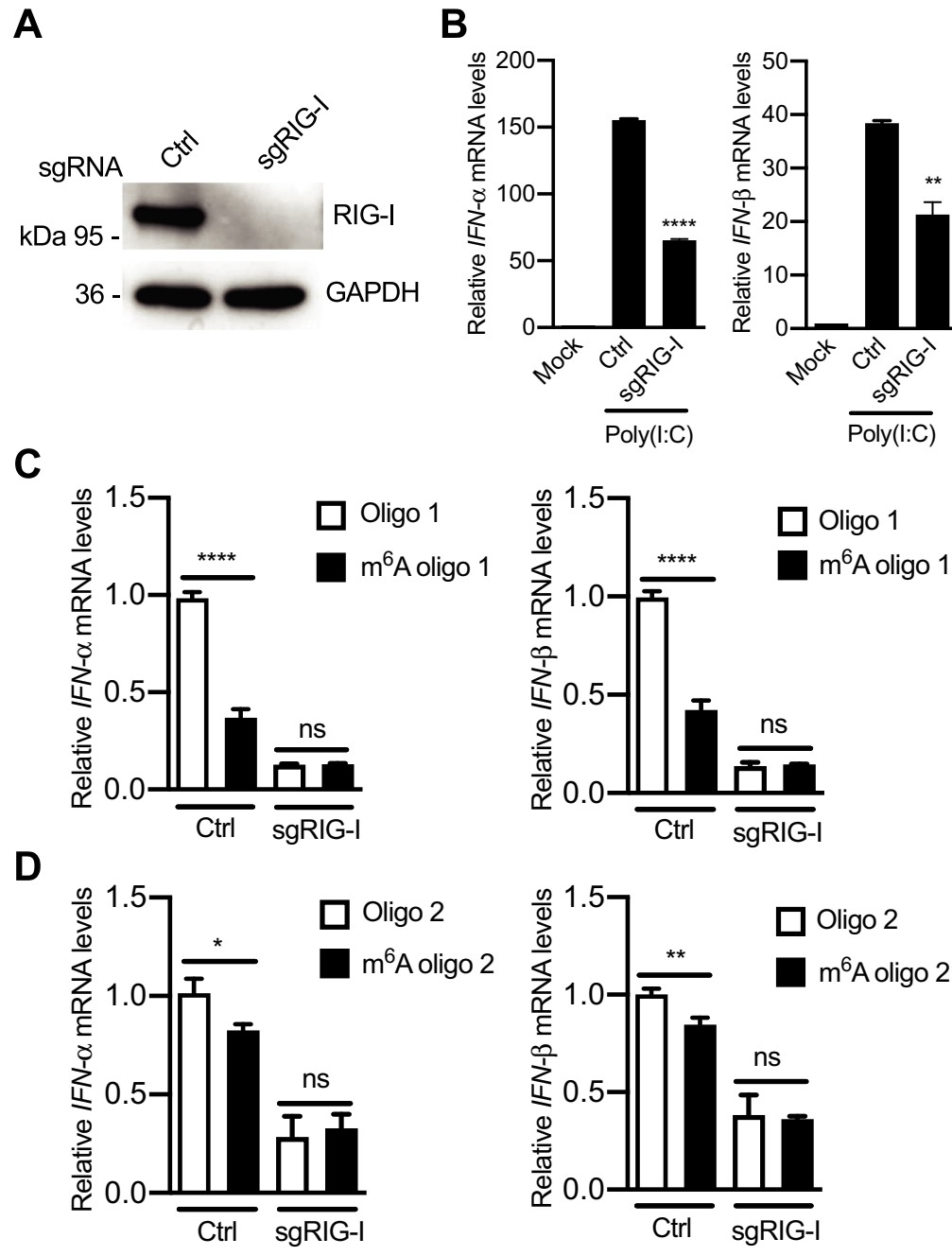

**Fig 10. RIG-I senses m$^6$A modification of HIV-1 RNA to induce IFN-I expression. (A)** RIG-I expression levels in control (Ctrl) and RIG-I knockout (sgRIG-I) U937 cells were measured by Western blotting. **(B)** Ctrl and RIG-I KO U937 cells were transfected with 250 ng of poly(I:C). At 16 hr post-transfection, cells were collected for the analysis of *IFN-α* and *IFN-β* mRNA levels by RT-qPCR. The results are shown as means ± S.D. of three repeats with similar results. $^{**} P < 0.005$, $^{****} P < 0.0001$. **(C and D)** PMA-differentiated Ctrl and RIG-I KO U937 cells were transfected with 250 ng of HIV-1 RNA oligo 1 **(C)** or oligo 2 **(D)**. After 16 hr, cells were collected for the analysis of *IFN-α* and *IFN-β* mRNA levels by RT-qPCR. The results are shown as means ± S.D. of three repeated experiments. $^{*} P < 0.05$, $^{**} P < 0.005$, $^{****} P < 0.0001$. Un-paired t-test was used for statistical analysis. ns, not significant.

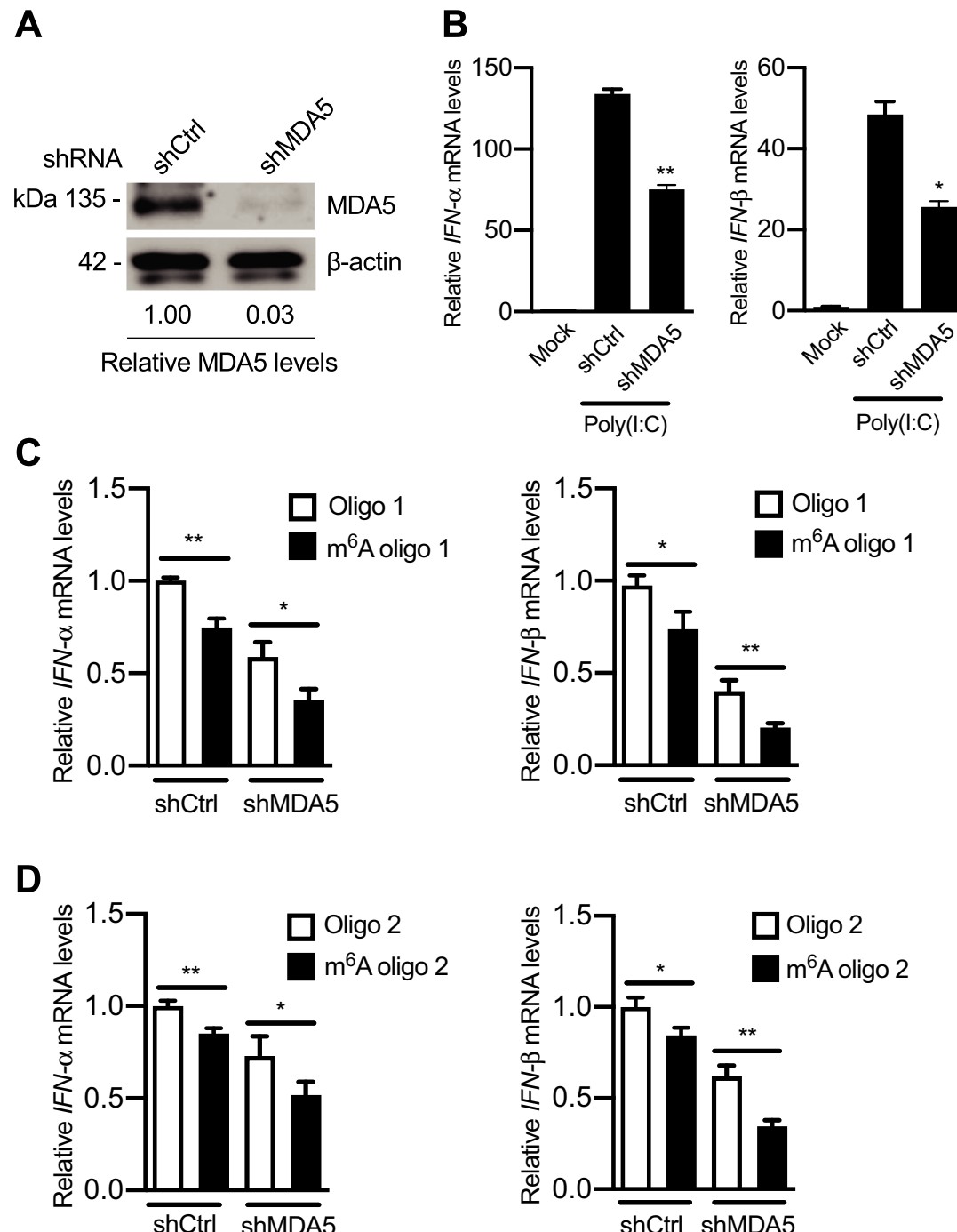

**Fig 11. MDA5 has no specific role in m⁶A modification of HIV-1 RNA to induce IFN-I expression. (A)** MDA5 expression levels were measured by Western blotting using control (shCtrl) and stable MDA5 knockdown (shMDA5) U937 cells. **(B)** shCtrl and shMDA5 U937 cells were transfected with poly(I:C). At 16 hr post-transfection, cells were collected for the analysis of *IFN-α* and *IFN-β* mRNA levels by RT-qPCR. The results are shown as means ± S.D. of three repeats with similar results. * $P < 0.05$, ** $P < 0.005$. **(C and D)** PMA-differentiated shCtrl and shMDA5 U937 cells were transfected with 250 ng of RNA oligo 1 **(C)** or oligo 2 **(D)**. At 16 h post-transfection, cells were collected for the analysis of *IFN-α* and *IFN-β* mRNA levels by RT-qPCR. * $P < 0.05$, ** $P < 0.005$. The results are shown as means ± S.D. of three repeated experiments.

showed that m$^6$A modifications of human metapneumovirus RNA mimic the host RNA to avoid RIG-I-mediated innate immune sensing, and thereby reduce the production of IFN-I and enhance viral replication [45]. However, it remains unknown whether m$^6$A modifications of HIV-1 RNA have any impact on innate immune responses.

In this study, we show that m$^6$A modifications of HIV-1 RNA act as a negative regulator of IFN-I induction by evading RNA sensing in PMA-differentiated U937 cells and primary macrophages from healthy blood donors. We observed that two different HIV-1 RNA oligos of the HIV-1 5′-UTR containing a single m$^6$A-modification significantly reduced IFN-I induction relative to their unmodified RNA counterparts. The different inhibitory effects on IFN-I induction by two m$^6$A-modified RNA oligos compared to their unmodified counterparts might be due to different sequences or conformation of the RNA fragments [21]. We also demonstrated that HIV-1 RNA with decreased m$^6$A levels enhanced IFN-I expression, but HIV-1 RNA with increased m$^6$A modifications had opposite effects. Our results suggest that HIV-1 genomic RNA and viral transcripts are masked by m$^6$A modifications to avoid RIG-I-mediated sensing and IFN-I induction during viral infection. Thus, HIV-1 has likely evolved an immune evasion strategy through m$^6$A modification of viral RNA in macrophages (Fig 12).

Several RNA modifications, such as N-1-methylpseudouridine, 5-methylcytidine (m$^5$C), 5-hydroxymethylcytidine, 5-methoxycytidine, and 2′ fluoro-deoxyribose, have a significant impact on RIG-I- and MDA5-mediated RNA sensing [46]. In addition to m$^6$A modification, HIV-1 genomic RNA contains eight types of epitranscriptomic modifications that are higher than the average cellular mRNA, with m$^5$C and 2′-O-methyl modifications being most prevalent [47]. It is possible that HIV-1 RNA exploits multiple epitranscriptomic modifications to avoid innate sensing as mechanisms of immune evasion.

The IFN-I gene itself is m$^6$A-modified and targets its destabilization for the maintenance of homeostatic state in mice and humans [44]. Virus infection of host cells can affect m$^6$A-modifications of cellular RNA and IFN-I responses. Rubio *et al.* showed that, following human cytomegalovirus infection, depletion of METTL14 or increase in ALKBH5 proteins leads to decrease the level of m$^6$A in IFN-β gene and stabilizes and elevates the IFN-I response [43]. We have reported that the HIV-1 envelope protein gp120 upregulates m$^6$A levels of cellular RNA in primary CD4$^+$ T cells or Jurkat cells independently of viral replication [23]. It is possible that HIV-1 may modulate the activity or localization of writers or erasers, thereby upregulating m$^6$A levels in HIV-1 infected cells. It remains to be established whether m$^6$A modifications of HIV-1 RNA regulate innate immune responses in primary CD4$^+$ T-cells. Moreover, recent studies suggest that HIV-1 intron-containing RNA activates innate immune signaling and IFN-I induction in primary dendritic cells, macrophages, microglia, and CD4$^+$ T cells [48–50]. It is conceivable that, during chronic HIV-1 infection, multiple cellular and viral mechanisms can regulate IFN-I induction and innate immune activation in infected cells and individuals.

Our data suggest that m$^6$A-modified HIV-1 reduces the activation of IRF3 and IRF7 through RIG-I-mediated signaling to suppress IFN-I induction. However, it remains unclear how m$^6$A modifications of HIV-1 RNA reduce phosphorylation of IRF3 and IRF7 during early stage of HIV-1 infection. Previous studies suggest that HIV-1 proteins can target several cellular RNA and DNA sensors including RIG-I to surpass the IFN-I response [51–54]. Moreover, HIV-1 can also target downstream proteins in the IFN-I pathway including IRF3 and IRF7 to contribute to chronic and persistent infection [55–59]. For example, HIV-1 Vpr protein mediates degradation of IRF3 to avoid the innate antiviral immune response [60].

Durbin *et al.* showed that a RIG-I-activating RNA ligand, the 106-nucleotide polyU/UC sequence derived from the 3′ UTR of hepatitis C virus with m$^6$A modification bound RIG-I with low affinity and did not trigger the conversion to the activated RIG-I conformer and thus

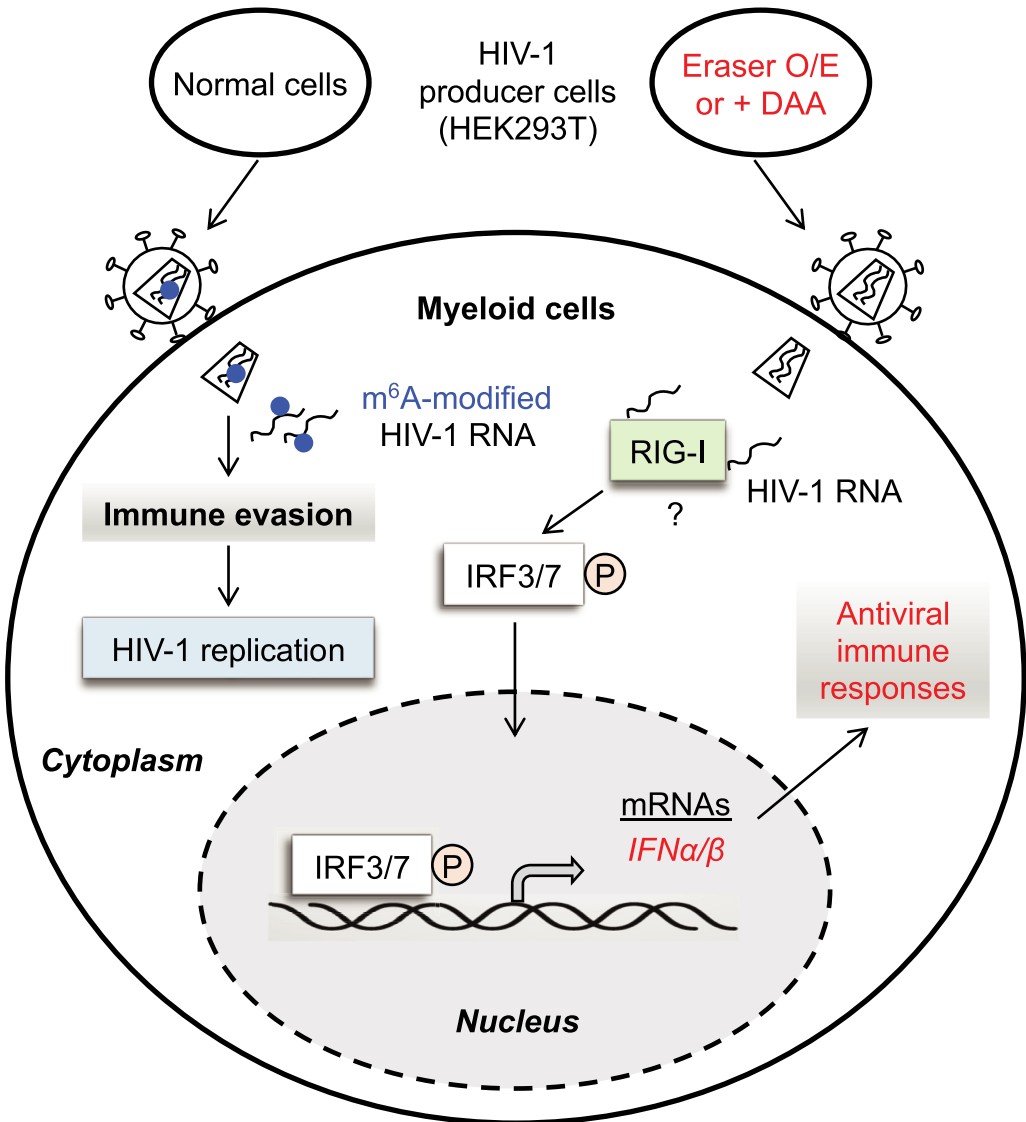

**Fig 12. HIV-1 RNA escapes from innate immune surveillance in myeloid cells.** In HIV-1 producer cells, writers add and erasers remove internal m⁶A modifications (blue dots) of viral RNA, respectively. HIV-1 with m⁶A-modified RNA avoids innate sensing in infected myeloid cells, thereby escaping immune surveillance. Overexpression (O/E) of erasers or inhibiting m⁶A addition with DAA in HIV-1 producer cells generates viruses with m⁶A-deficient viral RNA. When HIV-1 with m⁶A-deficient RNA infects myeloid cells including macrophages, the cytoplasmic RNA sensor RIG-I recognizes unmodified HIV-1 RNA and triggers phosphorylation (indicated by the letter P) of the transcription factors IRF3 and IRF7. Phosphorylation of IRF3/7 leads to IFN-α/β expression and generates antiviral innate immune responses in HIV-1-infected macrophage-like cells. However, it remains to be established whether m⁶A-deficient HIV-1 RNA enhances binding to RIG-I, thereby inducing IRF3/7 activation and IFN-I expression in cells.

has an immunosuppressive potential [46]. Our data indicated that m⁶A-deficient HIV-1 RNA oligos enhanced RIG-I-mediated RNA sensing and IFN-I induction in cells, while this effect could be independent of RNA sequence because an m⁶A-deficient RNA oligo with a random sequence also induced higher IFN-I mRNA compared to the unmethylated counterpart. Further studies are needed to examine whether the m⁶A-modified HIV-1 RNA binds RIG-I with a low affinity, which might be the possible cause of reduced IFN-I induction during viral infection.

In summary, our study uncovered a previously unidentified strategy of how HIV-1 RNA escapes the host antiviral innate immune system through m6A modifications of its RNA genome. HIV-1 RNA m6A modifications can act as an immune suppressor of RIG-I-mediated viral RNA sensing in myeloid cells. Our findings also implicate that pharmacological reduction in m6A modification of HIV-1 RNA could enhance IFN-I-mediated innate antiviral immune responses.

## Materials and methods

### Ethics statement

The Institutional Review Board (IRB) at the University of Iowa has approved the *in vitro* experiments in this study involving human blood cells from de-identified healthy donors. The consent requirements for the de-identified blood samples were waived by IRB.

### Cell culture

HEK293T cell line was a kind gift from Vineet KewalRamani (National Cancer Institute, USA) and maintained in complete Dulbecco's modified Eagle's medium (DMEM) as described [21]. U937 cell line was obtained from the American Type Culture Collection (ATCC) and maintained in complete RPMI-1640 medium as described [61]. All the cell lines were maintained at 37°C in 5% $CO_2$ and tested negative for mycoplasma contamination using a universal mycoplasma detection kit (ATCC 30-1012K) as described [42].

### Preparation of primary MDM

CD14+ primary monocytes were isolated with Anti-Human CD14 Magnetic Particles-DM (BD Biosciences, catalog no. 557769) from healthy donors' blood samples purchased from the DeGowin Blood Center at the University of Iowa. MDMs were differentiated from CD14+ monocytes as described [42].

### Plasmids and HIV-1 RNA oligos

The HIV-1 proviral DNA construct pNL4-3 was used to generate viral stocks as described [20]. For over-expression of the m6A erasers, the corresponding control vectors, pCMV6-FTO, pCMV-ALKBH5 were described [9, 62]. For knockout of eraser genes, CRISPR-Cas9 vectors containing sgControl, sgFTO, and sgALKBH5 were used as described [44]. For RIG-I knockout, pCR-BluntII--Topo-sgRIGI-1 and 2 vectors were described [63], which were kindly provided by Dr. Stacy Horner (Duke University, USA), and the plasmid hCas9 (catalog no. 41815, Addgene) was described [64]. For MDA5 and RIG-I knockdown, shControl, shMDA5 and shRIG-I plasmids [37] were kindly provided by Dr. Yamina Bennasser (Université de Montpellier, France). Six RNA oligo sequences are from the 5′ UTR of HIV-1 genomic RNA (NL4-3 strain) with or without a single m6A site [21], which were commercially synthesized (Integrated DNA Technologies, IDT). The sequences and the location of the m6A sites in the conserved GGACU motifs of the HIV-1 genome were described [21] and are listed below: RNA oligo 1 (nt. 235–281, the m6A-modified adenosine is nt. 241):

5′-CGCAGGACUCGGCUUGCUGGAGACGGCAAGAGGCGAGGGGCG-3′.

To eliminate RNA dimerization in our previous RNA binding assays [21], the original dimer initiation sequence of HIV-1 (AAGCGCGC) in oligo 1 was replaced with the underlined nucleotides GAG. RNA oligo 2 (nt. 176–217, the m6A-modified adenosine is nt. 197):

5′-AGCAGUGGCGCCCGAACAGGGACUUGAAAGCGAAAGUAAAGC-3′. The sequence of scrambled RNA oligo 2 was obtained using the Sequence Manipulation Suite (Bioinformatics. org). The m6A-modified adenosine of scrambled RNA oligo 2 is underlined and in bold as follow: 5′-AAAGGCGGUGCCAACCGCGAGGCUGGAGACAUAAAAACGAGU-3′.

## Generation of U937 cells with MDA5 knockdown or RIG-I knockout, and HEK293T cells with FTO or ALKBH5 knockout

For MDA5 knockdown U937 cell line construction, HEK293T cells were transfected with shControl or shMDA5, together with pMD2.G and psPAX2 plasmids by polyethyleneimine (PEI) [42]. At 48 hr post-transfection, lentiviruses were harvested and purified to infect U937 cells for 48 hr and then the U937 cells were selected in RPMI-1640 media with 1 μg/mL puromycin. To generate RIG-I knockout cells, pCR-BluntII-Topo-sgRIGI-1 or pCR-BluntII-Topos-gRIGI-2, along with hCas9, which has neomycin (G148) resistance, were transfected into U937 cells by TransIT mRNA transfection kit (Mirus, USA) for 48 hr according to the manufacturer's protocol. Then, G418 (1 mg/mL) was added to transfected cells for 8 days to select RIG-I knockout U937 cells, which were confirmed by Western blotting. For Control, FTO, and ALKBH5 knockout HEK293T cell generation, HEK293T cells were transfected with corresponding single guide RNAs (sgRNAs), together with pMD2.G and psPAX2 plasmids. At 48 hr post-transfection, lentiviruses were collected to infect fresh HEK293T cells for 48 hr. Then, the single clones were selected by 1 μg/mL puromycin in 96 well plates. The KO cells were confirmed by DNA sequencing and for specific protein expression by Western blotting.

## Generation of m⁶A-reduced HIV-1 from cells treated with DAA

HEK293T cells were mock-treated with PBS or 50 μM DAA (Sigma-Aldrich, Product number D8296) for 4 hr before transfection with pNL4-3 plasmid as described [19, 20]. DAA concentration (50 μM) was maintained in the culture medium for an additional 48 hr. Cells and supernatants were collected at 48 hr post-transfection for the analysis of protein expression and RNA m⁶A by immunoblotting.

## Dot immunoblotting of m⁶A modification in RNA

RNA was extracted from purified and concentrated HIV-1 stocks by using TRIzol (Invitrogen) or RNA purification kit (Qiagen). The synthesized RNA oligos were directly used for dot-blot assays as described [23]. Briefly, HIV-1 RNA or RNA oligos (diluted to 100 μL using 1 mM EDTA) were mixed with 60 μL of 20× saline-sodium citrate (SSC) buffer (3 M NaCl, 0.3 M trisodium citrate) and 40 μL of 37% formaldehyde (Invitrogen) and incubated at 65°C for 30 min. Nitrocellulose membrane (Bio-Rad, catalog no. 162–0115) or nylon membranes (Roche, catalog no. 11209299001) were pre-soaked with 10X SSC for 5 min and assembled in dot-blot apparatus (Bio-Rad) with vacuum-on. Equal amounts of RNA were transferred to nitrocellulose or nylon membranes, then membranes were washed twice with 200 μL of 10× SSC buffer. Nylon membranes were washed once with TBST buffer (20 mM Tris, 0.9% NaCl, and 0.05% Tween 20) for 5 min and stained with methylene blue staining (Molecular Research Center, catalog no. MB119) for 2–5 sec followed by two or three washes with ddH₂O. Nitrocellulose membranes were blocked with 5% milk in TBST buffer and used to detect m⁶A levels by probing with an m⁶A-specific antibody (Synaptic Systems, catalog no. 202 003). Images were taken by Amersham Biosciences Imager 600 (GE Healthcare) and analyzed by ImageJ software (National Institutes of Health). Densitometry quantification of relative RNA m⁶A levels was normalized to MB staining as described [23].

## *In vitro* FTO demethylation of HIV-1 RNA m⁶A

Demethylation of HIV-1 RNA m⁶A was performed with recombinant FTO treatment of purified HIV-1 RNA. Briefly, 500 ng HIV-1 RNA were used for FTO *in vitro* treatment in 100 μL reaction buffer containing 50 mM HEPES buffer (pH7.0), 75 μM (NH4)$_2$Fe (SO4)$_2$•6H$_2$O, 2

mM L-ascorbic acid, 300 μM 2-KG (alpha-ketoglutaric acid), 200 U RNasin ribonuclease inhibitor, 5 μg/mL BSA, and 0.2 nmol FTO protein. The reaction was performed at 37˚C for 1 hr and then stopped by adding 5 mM EDTA. Finally, RNA samples were denatured at 70˚C for 2 min and quickly put into ice for m$^6$A detection.

### HIV-1 production, p24 quantification, RNA transfection of U937 cells and MDM, and HIV-1 infection assays

HIV-1 stocks were generated by transfection of HEK293T cells with the proviral DNA pNL4-3 using PEI as described [42]. Cell culture medium was exchanged at 6–8 hr post-transfection with supernatants and was harvested at 48 hr. The cell culture media containing viruses were filtered (0.45 μm) and purified by 25% sucrose using an SW28 rotor (Beckman Coulter) at 141,000$g$ for 90 min. The pellet was resuspended with PBS and digested with DNase I (Turbo, Invitrogen) for 30 min at 37˚C. To extract HIV-1 genome RNA, concentrated HIV-1 virions were lysed by Trizol (Invitrogen) and RNA was purified by phenolic-chloroform sedimentation and isopropanol precipitation.

For RNA transfection, U937 cells were treated with 100 ng/mL phorbol 12-myristate 13-acetate (PMA) for 24 hr and changed with fresh RPMI-1640 media for another 24 hr. PMA-differentiated U937 cells ($5 \times 10^5$) were then transfected with TransIT mRNA transfection kits (Mirus) according to the manufacturer protocol. Transfection of poly(I:C) (Sigma-Aldrich, P1530) was used as a positive control. At 16 hr post-transfection, cells were harvest for RT-qPCR analysis. MDM ($5 \times 10^5$) were transfected with isolated HIV-1 RNA (125 ng) or GFP-encoding mRNA (1 μg, a kind gift from Dr. Zack Zhou at the Luna Innovations Inc.) using the TransIT-mRNA transfection kit (Mirus). At 24 hr post-transfection, *IFN-I* mRNA and GFP expression in MDM was measured by RT-qPCR and by flow cytometry, respectively.

For HIV-1 infection assays, HIV-1 p24 levels were quantified by an enzyme-linked immunosorbent assay (ELISA) using anti-p24-coated plates (The AIDS and Cancer Virus Program, NCI-Frederick, MD) as described [23]. PMA-differentiated U937 cells ($5 \times 10^5$) were infected by equal amounts of HIV-1 (250 pg of p24) for 16 h and then cells were collected for Western blotting or RT-qPCR analysis.

### Measurement of HIV-1 infectivity

To determinate the infectivity of HIV-1$_{NL4-3}$, an equivalent amount of virus stock (62.5, 125, or 250 pg of p24) was used to infect TZM-bl cells [38, 39] in 48-well plates in the presence of DEAE-dextran (40 μg/ml, Sigma-Aldrich). At 48 hpi, TZM-bl cells were washed twice with PBS and lysed for the luciferase assay (Promega) following the manufacturer's instructions. Cell protein concentrations were quantified using a bicinchoninic acid assay (Pierce) and all luciferase results were normalized based on total protein input.

### Antibodies and immunoblotting

The antibodies used in this study were: anti-GAPDH (AHP1628, Bio-Rad), anti-FLAG (F1804, Sigma-Aldrich), anti-METTL3 (15073-1-AP, Proteintech Group), anti-METTL14 (HPA038002, Sigma-Aldrich), anti-FTO (ab124892, Abcam), anti-ALKBH5 (HPA007196, Sigma-Aldrich), anti-MDA5 (D74E4, Cell Signaling), anti-RIG-I (D14G6, Cell Signaling), anti-HIV-1 Gag (clone #24–2, the NIH AIDS Reagent Program), anti-IRF3 (124399, Abcam), anti-phospho-IRF3 (Ser396) (4D4G) (4947, Cell Signaling), anti-IRF7 (4920S, Cell Signaling), anti-phospho-IRF7 (Ser471/472) (5184, Cell Signaling) and anti-m$^6$A polyclonal rabbit antibody (202 003, Synaptic Systems). Cells were harvested and lysed in cell lysis buffer (Cell Signaling) supplemented with a protease inhibitor cocktail (Sigma-Aldrich) and a phosphatase

inhibitor cocktail (Cell Signaling). Immunoblotting was performed as described [23]. Detection of GAPDH expression was used as a loading control.

## IFN-I protein detection by ELISA

The concentrations of released IFN-α and IFN-β proteins in the supernatants of MDM transfected with HIV-1 RNA or mock-transfected at 24 hr post-transfection were detected by Veri-Kine ELISA kits for human IFN-α (catalog no. 41100) and IFN-β (catalog no. 41410) according to the product instructions (PBL Assay Science).

## Quantitative RT-PCR

Real-time quantitative RT-PCR (qRT-PCR) was performed as described [42, 65] to assess the relative levels of *IFN-α*, *IFN-β*, HIV-1 *gag* mRNA expression in cells induced by HIV-1 RNA transfection or HIV-1 infection. Following primers (IDT) were used:

IFN-α, F 5'-GTACTGCAGAATCTCTCCTTTCTCCT-3'
IFN-α, R 5'-GTGTCTAGATCTGACAACCTCCCAGG-3'
IFN-β, F 5'-AACTTTGACATCCCTGAGGAGATTAAGC-3'
IFN-β, R 5'-GACTATGGTCCAGGCACAGTGACTGTAC-3'
Gag-F, 5'-CTAGAACGATTCGCAGTTAATCCT-3'
Gag-R, 5'-CTATCCTTTGATGCACACAATAGA G-3'
GAPDH, F 5'-GGAAGGTGAAGGTCGGAGTCAACGG-3'
GAPDH, R 5'-CTGTTGTCATACTTCTCATGGTTCAC-3'

## Statistical analyses

Data were analyzed using either Mann-Whitney's t-test or the one-way or two-way analysis of variance (ANOVA) with Prism software and statistical significance was defined as $P < 0.05$. All experiments were repeated 2–3 times as indicated in the figure legends.

## Acknowledgments

We thank Jack T. Stapleton, Jennifer Welch, Alexis Hawkins and the Wu lab members for helpful discussions and suggestions. The authors appreciate generous reagents from Drs. Yamina Bennasser, Ruiqi Ge, Stacy Horner, Vineet KewalRamani, Noam Stern-Ginossar, Zack Zhou, and the NIH AIDS Reagent Program.

## Author Contributions

**Conceptualization:** Chuan He, Li Wu.

**Data curation:** Shuliang Chen, Sameer Kumar, Constanza E. Espada, Nagaraja Tirumuru, Michael P. Cahill.

**Formal analysis:** Shuliang Chen, Sameer Kumar, Constanza E. Espada, Nagaraja Tirumuru, Li Wu.

**Funding acquisition:** Chuan He, Li Wu.

**Investigation:** Shuliang Chen, Sameer Kumar, Constanza E. Espada, Nagaraja Tirumuru, Li Wu.

**Methodology:** Shuliang Chen, Sameer Kumar, Constanza E. Espada, Lulu Hu.

**Project administration:** Li Wu.

**Resources:** Lulu Hu, Chuan He, Li Wu.

**Supervision:** Li Wu.

**Validation:** Li Wu.

**Writing – original draft:** Shuliang Chen, Sameer Kumar, Li Wu.

**Writing – review & editing:** Shuliang Chen, Sameer Kumar, Constanza E. Espada, Nagaraja Tirumuru, Chuan He, Li Wu.

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
