## [Decision Letter · Decision Letter 0]

16 Nov 2020

Dear Dr. Wu,

Thank you very much for submitting your manuscript "N6-methyladenosine modification of HIV-1 RNA evades RIG-I-mediated sensing to suppresses type-I interferon induction in monocytic cells" for consideration at PLOS Pathogens. As with all papers reviewed by the journal, your manuscript was reviewed by members of the editorial board and by several independent reviewers. In light of the reviews (below this email), we would like to invite the resubmission of a significantly-revised version that takes into account the reviewers' comments. We cannot make any decision about publication until we have seen the revised manuscript and your response to the reviewers' comments. Your revised manuscript is also likely to be sent to reviewers for further evaluation.

Sincerely,

David T. Evans

Associate Editor

PLOS Pathogens

Thomas Hope

Section Editor

PLOS Pathogens

Kasturi Haldar

Editor-in-Chief

PLOS Pathogens

orcid.org/0000-0001-5065-158X

Michael Malim

Editor-in-Chief

PLOS Pathogens

orcid.org/0000-0002-7699-2064

Reviewer's Responses to Questions

**Part I - Summary**

Reviewer #1: In this original research article, Chen et al. look to test the hypothesis that m6A modification of the HIV RNA genome modulate the innate immune responses by inhibition of RNA sensing. Using a combination of in vitro RNA modifications and ex vivo gene editing approaches to alter m6A pathways, the authors show a very small, but consistent effect where less m6A correlates with more IFN induction in the monocytic cell line U937 and vice versa. Knockout of the sensors RIG-I and MDA5 suggest a reliance on RIG-I for sensing. Perhaps paradoxically to their argument, they find decreased levels of m6A modification in cellular RNA from virally suppressed patients on ART compared to viremic controls correlating with decreased IFN levels in vivo.

While the paper is generally well-written, with a clear hypothesis and a good attempt to bridge ex vivo and in vivo experimentation, the overall magnitude of the effect is not convincing, the reliance on one cell line model is limiting, the final mechanistic model is unclear, and several controls are missing that would improve interpretation of the results. Overall, while the results are consistent with a model in which m6A modification to RNA may provide limited protection against RIG-I sensing, it is unclear to what extent, if any, this is relevant to HIV infection. No effects on HIV infectivity are shown and the data collected in vivo seem at odds to their in vitro model. Several additional experiments are required to support the claims made in the paper.

Reviewer #2: In this study, the authors showed that m6A modification of HIV-1 RNA suppresses the expression of IFN-I in human monocytic cell line U937. The authors demonstrated this effect of m6A via 1) transfection of cells with 42-mer RNA oligos (corresponding to sequences in HIV 5’UTR) that contain (or not) single m6A modification, 2) transfection of cells with RNA purified from virions with varying degrees of the m6A modification, and 3) infection of cells with HIV-1 containing different levels of the m6A modification. The authors further presented the data suggesting that the m6A modification allows HIV-1 to reduce activation of the transcription factors IRF3 and IRF7 that drive IFN-I gene expression and that m6A modification allows viral RNA to evade sensing by RIG-I but not MDA5. While the effects of changes in the m6A modification are modest in some experiments, the use of multiple methods to alter m6A modification levels makes the data convincing. These data are clearly presented and interpreted for the most part in a balanced manner. As the finding that m6A modification of HIV-1 RNA promotes virus evasion of innate immune sensing is novel and potentially important, the manuscript is likely of interest to many readers of PLOS Pathogens. There are, however, a few points that the authors need to consider to improve the manuscript as described below.

Reviewer #3: Summary:

Chen and colleagues study the impact of HIV RNA m6A modifications on the induction of innate immune responses in monocytic cells. It has been recently found that m6A modifications can have an impact on RIG-I mediated infection in metapneumovirus, but similar studies have not been conducted with regards to HIV-1. The understanding of HIV-1 interactions with innate sensing machinery is very important to understanding HIV-1 pathogenesis, so this study is of high interest to both HIV researchers and general viral pathogenesis researchers. Using macrophage systems, the authors demonstrate the impact that m6A modification of the HIV-1 RNA can have on innate sensing. Transfection of differentiated monocytes with oligos harboring an m6A modification resulted in significantly reduced INF-alpha and –beta mRNA levels when compared to transfections using the same oligos without RNA alteration. The authors further support their hypothesis using a knockout of two different m6A eraser molecules driving a decrease in IFN-I responses. Interestingly, the same was observed when m6A was specifically inhibited by the compound DAA. In contrast, overexpression of the same erasers lead to the opposite effect. In addition, Chen et al. showed that infection with HIV RNA leads to phosphorylation/activation of IRF3 and IRF7, key regulators of the IFN-I pathways. The effect was even more pronounced when using conditions that increase m6A levels. In line with this, knockout of the RIG-I complex, a major component of the same pathway caused a significant decrease in IFN-I responses. Finally, the authors compared m6A levels and IFN-I responses in two HIV patient groups (viremic versus ART treatment) with a healthy control group. Both, m6A and IFN-I levels were highest in the viremic group and lowest in ART treated HIV patients.

Overall the quality of the data are high and the authors are careful to correlate the extent of m6A modification with the magnitude of the effects that they see. They present a robust data set in support of their hypothesis and their conclusions are sound. The size the effects that they measure are modest, but clearly affect IFN-I mRNA levels and IRF phosphorylation. It may be noted that the magnitude of the effect is weaker than strong agonists like poly IC, so it may be helpful to determine how vigorously this translates into differences in IFN-I protein levels or bioassay. Also although the impact of IFN-I on macrophage replication may not be expected to be large, it is worth commenting on whether this level of IFN-I induction has any affect on viral replication in monocytes (as I suspect this phenotype is more relevant to inflammation rather than direct effects on replication.) Lastly the patient data are intriguing, though an explanation for how the increase in m6A in patients relates to the monocyte cellular models, which indicate that elevated m6A should work to decrease inflammation, whilst it is clear that in vivo infection is associated with elevated IFN-I. This paper is of high interest.

**Part II – Major Issues: Key Experiments Required for Acceptance**

Reviewer #1: 1) Figure 1: While the oligo assay is clean and convenient, the relationship to HIV is tentative at best. While the 42-mers match HIV sequences, they are no substitute for incoming viral RNA or viral particles (tested directly in later figures). Rather, this figure seems to indicate a small (0.2 – 4.0 fold) effect on IFN induction by m6A modification in a sequence dependent manner. As the data is only shown normalized and without any positive or negative induction controls, it is unclear to what extent the oligos induce a response over baseline. Does m6A modification protect other oligos in random sequence? Would this difference be seen in other cell line models or in primary macrophages?

2) Figure 2: In vitro modification and subsequent transfection of HIV RNA results in 10-fold changes to m6A levels, but this has barely a 3-fold effect on IFN. HIV RNA from FTO overexpression cells similarly has 10-fold less m6A, but barely results in a 2-fold effect on IFN compared to a 200-fold effect of the positive control. Is there any proposed explanation for the differential scaling of these effects? Could this be driven by changes to RNA stability in the cell after delivery, which m6A has been shown to directly effect, rather than by direct protection from sensing?

3) Figure 2: This is further examined by infection with viruses produced from FTO overexpression cells and these show again a barely 2-fold increase in IFN over control viruses. Would a two-fold effect on IFN influence HIV replication overall? What are the percent infected cells after this challenge was performed?

4) Figure 2: The overall model that these data suggest is also somewhat puzzling as, to be sensed, the genomic RNA has to be released from the core at some point. Have the authors tried doing this experiment in the presence of an RT inhibitor to show that this is modified genomic RNA and not abortive RT transcripts driving this response?

5) Figure 3: Similar concerns with scale are raised as with Figure 2. For example, ALKBH5 overexpression decreases m6A levels by 50%, but has a similar impact on IFN as FTO overexpression, which reduces m6A levels by 90%? This would suggest that the impact on IFN does not directly correlate with m6A levels, but may be limited by some other factor.

6) Figure 4: Knock-out of FTO and ALKBH5 look great, though the impact of ALKBH5 knock-out on HIV RNA m6A levels is greater than FTO, directly opposite the observation with overexpression. Transfection of this RNA shows a consistent though again minimal impact with up to 25-fold more m6A resulting in only a roughly 75% decrease in IFN induction. Parallel infection experiments show less than a 50% effect. This would be more convincing if the result could be rescued by treatment of the FTO-KO or ALKBH5-KO RNA with FTO in vitro. Still, the impact of m6A levels on RNA stability may serve as a significant confounder to these results.

7) Figure 5: The inhibitor is effective, but the IRF3 and IRF7 blots are nowhere near clear enough for precise quantification. The 1.2 and 1.7-fold effects are not convincing by these data.

8) Figures 1-5: At some point, the most pertinent results should be validated in primary cells.

9) Figures 6 and 7: Though the phenotype is still small, the RIG-I dependency (rather than MDA5) is clear. It would be better to compare two KO lines as the shMDA5 line may have residual activity. These results are consistent with previous reports from Durbin et al. that m6A specifically protects against RIG-I sensing. My biggest concern here is again it is unclear how much these small oligos really reflect sensing of HIV RNA, nonetheless HIV RNA in the context of infection. One small note to the authors is they may want to validate these lines with Sendai virus infection as it is specifically sensed by RIG-I, but not as much by MDA5.

10) Figure 8: While this effort to gather in vivo data is appreciated, it is unclear how monitoring overall levels of cellular RNA m6A in control, viremic, and ART patients contributes to the overall narrative of the manuscript. While the data show statistically significant differences in m6A levels between viremic and ART patients, the number of patients is too small to make any definitive conclusions or to meaningfully control for any demographic or virologic confounders. IFN levels in viremic individuals is higher than control or ART patients, consistent with prior reports, but this almost contradicts the model that more m6A protects against IFN induction.

Reviewer #2: 1. The patient-based data shown in Fig 8 are unrelated to the rest of the study described in the manuscript. This figure compares the average m6A level in total RNA of PBMCs from HIV-1 viremic patients with that from patients on ART. As the main focus of the manuscript is m6A modification of HIV-1 RNA, not the total cellular RNA, this part is irrelevant to the overall conclusion of the manuscript in the current form and thus should be removed.

2. The conclusion that m6A modification allows viral RNA to evade RNA sensing by RIG-I is based on the results obtained with transfection of the 42-mer RNA oligos (Figs 6 and 7). This should be confirmed through infection of cells with HIV-1 containing different levels of m6A modification as done in Figs 2-5.

3. All presented data (except for Fig 8) were obtained using a monocytic cell line U937. It would be ideal to confirm at least key parts of the results using primary monocytes or monocyte-derived APCs.

Reviewer #3: Major issues:

1) In differentiated monocytes the authors show that transfection/infection of HIV RNA with more m6A inversely correlates with IFN-I responses. They also come up with a model describing that m6A seems to mask HIV RNA from being recognized by the RNA sensing pathway in the host cell (Sup. Figure S2). However, in patients they show a positive correlation between m6A levels and IFN-I responses (the viremic group has highest levels of m6A and IFN-I). How does this fit into their in vitro model? This seems inconsistent with their in vitro model.

2) In their study, Chen et al measured all their IFN-I responses on RNA level. These also correlated with IRF phosphorylation, so this is promising. To demonstrate that there is an actual increase on the IFN protein level and/or a bioassay would be further compelling to demonstrate functional significance of the IFN-I mRNA levels.

3) In the last line of the discussion the authors suggest that antagonism of m6A modification can lead to increased IFN-I and decreased replication. Does it have an impact on replication? A discussion of how IFN-I may be expected to influence replication would help put this into context.

**Part III – Minor Issues: Editorial and Data Presentation Modifications**

Reviewer #1: (No Response)

Reviewer #2: (No Response)

Reviewer #3: Minor issues:

1) Page 10/11, lines 225-234: As a matter of interpretation: The authors describe in the MDA5 knockdown experiments with subsequent HIV RNA transfection/infection that they don’t see a full abrogated IFN-I response (Figure 5) as with knockdown of RIG-I and draw the conclusion that MDA5 does not contribute to the recognition of HIV RNA. Is it possible that the difference in m6A vs. non m6A induced IFN-I response in the knockdown experiment is due to the residual MDA5? Figure 7A shows that there is not a total knockdown of MDA5 as compared to RIG-I.

2) Page 11, line 245: There is no inverse correlation, but a positive correlation of m6A with viral load.

3) Page 12, line 256: There is no significant decrease comparing the HIV viremic and ART group for IFN-alpha, only for IFN-beta.

4) Figure 8A-C: Is there a correlation of viral load with m6A levels/IFN-alpha/IFN-beta levels within the viremic HIV group? A separate graph showing a correlation would be helpful.

5) We would suggest incorporating Sup Fig. S2 into the text.

6) Page 14, line 302: mice

7) Page 14, line 312: modifications

8) Page 14, line 316: reduce

PLOS authors have the option to publish the peer review history of their article (what does this mean?). If published, this will include your full peer review and any attached files.

Reviewer #1: No

Reviewer #2: No

Reviewer #3: No
---

## [Decision Letter · Decision Letter 1]

12 Feb 2021

Dear Dr. Wu,

Thank you very much for submitting your manuscript "N6-methyladenosine modification of HIV-1 RNA suppresses type-I interferon induction in differentiated monocytic cells and primary macrophages" for consideration at PLOS Pathogens. As with all papers reviewed by the journal, your manuscript was reviewed by members of the editorial board and by several independent reviewers. The reviewers appreciated the attention to an important topic. Based on the reviews, we are likely to accept this manuscript for publication, providing that you modify the manuscript according to the review recommendations.

Although all of the reviewers felt that your revised manuscript is greatly improved, Review 1 still has a few remaining concerns. As an opportunity to use these comments to further improve your study, I would ask that you respond to them to the extent that you feel is constructive.

Sincerely,

David T. Evans

Associate Editor

PLOS Pathogens

Thomas Hope

Section Editor

PLOS Pathogens

Kasturi Haldar

Editor-in-Chief

PLOS Pathogens

orcid.org/0000-0001-5065-158X

Michael Malim

Editor-in-Chief

PLOS Pathogens

orcid.org/0000-0002-7699-2064

Although all of the reviewers felt that your revised manuscript is greatly improved, Review 1 still has a few remaining concerns. As an opportunity to use these comments to further improve your study, I would ask that you respond to them to the extent that you feel is constructive.

Reviewer Comments (if any, and for reference):

Reviewer's Responses to Questions

**Part I - Summary**

Reviewer #1: In this original research article, Chen et al. look to test the hypothesis that m6A modification of the HIV RNA genome modulates the innate immune response to infection by inhibition of RNA sensing. Using a combination of in vitro RNA modifications and ex vivo gene editing approaches to alter m6A pathways, the authors show a small, but consistent effect where less m6A correlates with more IFN induction in the monocytic cell line U937 and vice versa. This phenotype is RIG-I dependent and further correlates with phosphorylation of IRF3/7. Important aspects of the phenotype are validated in primary MDMs.

This manuscript reflects a major improvement over the first submission. I would like to commend the authors for their hard work in responding to my comments, which I know were extensive. A majority of my concerns were addressed. The disconnect between the magnitude of the changes in m6A levels versus the magnitude of effect on IFN induction is never really addressed, but I can accept that this is something for future work. However, there is still one critical control that I think must be addressed prior to publication as it directly influences how these data are interpreted, namely viral RNA quantification before challenge (see Major Issues). Besides that, I only have a couple of minor suggestions.

Reviewer #2: In this revision, the authors added a substantial amount of new data, which provided supporting evidence for the physiological significance of the original observations and further advanced mechanistic understanding. Removing the in vivo data has also helped clarify the focus of the study. Overall, in my opinion, the authors have satisfactorily addressed the concerns raised by the reviewers in the previous round.

Reviewer #3: A revised manuscript from Chen et al has made extensive changes to address the questions raised in the initial review. The study is thorough in looking at the activation of type I IFN responses by RNA that is hyper or hypo modified by N6-methyladenosine. The most significant changes in the manuscript are the removal of the clinical data, which did not connect well with the observations made in cell lines. And the addition of very relevant data set regarding the response of primary monocyte derived macrophages (MDM) to changes in m6A. The studies in MDM show robust effects and are considerably more relevant to normal physiology of HIV and the induction of inflammatory responses. The authors find that m6A modification suppresses IFN induction in primary MDM and also find that strong enhancement of IFN induction when erasers are used to remove m6A modification. The modifications have strengthened the manuscript substantially.

**Part II – Major Issues: Key Experiments Required for Acceptance**

Reviewer #1: Figures 3-8 all rely on an assay wherein HIV-1 is harvested from culture supernatants and subsequently used for downstream challenge, either directly or using extracted RNA. In most cases, the authors show that the cells express similar levels of p24, that different conditions have similar levels of total RNA, and that the expected changes in m6A levels are observed. A critical missing component, however, is how much live virus or viral RNA is contained in those samples. As the changes in IFN induction are only 2-3 fold in most cases, small differences in the amount of virus produced or the amount of viral RNA packaged could drastically influence the downstream result. Given the new data in Figure 4B (that there is 30-fold more HIV-1 gag RNA produced after challenge of U937 cells with virus from FTO o/e cells), this information is even more critical. Is altering FTO or m6A pathways in producer cells influencing downstream IFN sensing in target cells primarily by influencing the amount of viral RNA produced or packaged? If you DNase treat the RNA in Figure 3C, for example, and do qPCR for viral RNA, is it equal? I think this is really important for interpretation of the phenotype (and may also speak to the magnitude issue mentioned above… I suspect that most of the RNA extracted is not actually viral RNA, but cellular RNA from exosomes isolated alongside the VLPs. All of this RNA may show changes in m6A levels, but it might not all be sensed like viral RNA is.).

Reviewer #2: I have no more concerns.

Reviewer #3: No major issues noted.

**Part III – Minor Issues: Editorial and Data Presentation Modifications**

Reviewer #1: 1) I would include a paragraph in the introduction talking more about previous molecular studies that show the importance of the m6A pathway in HIV-1 replication.

2) Figure 8 shows minimal impacts on m6A after FTO-OE, but huge impacts on sensing. If only some sites in the genome are important for sensing, m6A-Seq may be very informative here. I know this is outside the scope, but it would be really cool to do.

3) I actually find the rescue data that you shared very compelling! As discussed above, I would bet that slight discrepancies in the amount of viral RNA may explain some of the fluctuation. If you have a chance to repeat it, it might trend towards a significant difference at which point I would highly recommend including it in the manuscript.

Reviewer #2: (No Response)

Reviewer #3: Might add a bit of introductory background regarding the role of monocytes in pathogen sensing and inflammation in HIV. This bit of additional context in the intro may add to the significance of the study.

PLOS authors have the option to publish the peer review history of their article (what does this mean?). If published, this will include your full peer review and any attached files.

Reviewer #1: No

Reviewer #2: No

Reviewer #3: No
---

## [Editor Report · Decision Letter 2]

25 Feb 2021

Dear Dr. Wu,

We are pleased to inform you that your manuscript 'N6-methyladenosine modification of HIV-1 RNA suppresses type-I interferon induction in differentiated monocytic cells and primary macrophages' has been provisionally accepted for publication in PLOS Pathogens.

Best regards,

David T. Evans

Associate Editor

PLOS Pathogens

Thomas Hope

Section Editor

PLOS Pathogens

Kasturi Haldar

Editor-in-Chief

PLOS Pathogens

orcid.org/0000-0001-5065-158X

Michael Malim

Editor-in-Chief

PLOS Pathogens

orcid.org/0000-0002-7699-2064
---

## [Editor Report · Acceptance letter]

5 Mar 2021

Dear Dr. Wu,

We are delighted to inform you that your manuscript, "N6-methyladenosine modification of HIV-1 RNA suppresses type-I interferon induction in differentiated monocytic cells and primary macrophages," has been formally accepted for publication in PLOS Pathogens.

Best regards,

Kasturi Haldar

Editor-in-Chief

PLOS Pathogens

orcid.org/0000-0001-5065-158X

Michael Malim

Editor-in-Chief

PLOS Pathogens

orcid.org/0000-0002-7699-2064